# LATENT INTUITIVE PHYSICS: LEARNING TO TRANSFER HIDDEN PHYSICS FROM A 3D VIDEO

**Xiangming Zhu**\* **Huayu Deng**\* **Haochen Yuan**\* **Yunbo Wang**† **Xiaokang Yang**
MoE Key Lab of Artificial Intelligence, AI Institute, Shanghai Jiao Tong University
{xmzhu76, deng_hy99, yuanhaochen, yunbow, xkyang}@sjtu.edu.cn
https://sites.google.com/view/latent-intuitive-physics/

## ABSTRACT

We introduce latent intuitive physics, a transfer learning framework for physics simulation that can infer hidden properties of fluids from a single 3D video and simulate the observed fluid in novel scenes. Our key insight is to use latent features drawn from a learnable prior distribution conditioned on the underlying particle states to capture the invisible and complex physical properties. To achieve this, we train a parametrized prior learner given visual observations to approximate the visual posterior of inverse graphics, and both the particle states and the visual posterior are obtained from a learned neural renderer. The converged prior learner is embedded in our probabilistic physics engine, allowing us to perform novel simulations on unseen geometries, boundaries, and dynamics without knowledge of the true physical parameters. We validate our model in three ways: (i) novel scene simulation with the learned visual-world physics, (ii) future prediction of the observed fluid dynamics, and (iii) supervised particle simulation. Our model demonstrates strong performance in all three tasks.

## 1 INTRODUCTION

Understanding the intricate dynamics of physical systems has been a fundamental pursuit of science and engineering. Recently, deep learning-based methods have shown considerable promise in simulating complex physical systems (Battaglia et al., 2016; Mrowca et al., 2018; Schenck & Fox, 2018; Li et al., 2019; Ummenhofer et al., 2020; Sanchez-Gonzalez et al., 2020; Shao et al., 2022; Prantl et al., 2022; Han et al., 2022; Guan et al., 2022; Li et al., 2023). However, most previous works focus on physics simulation with given accurate physical properties, which requires strong domain knowledge or highly specialized devices. *Let us consider a question: Can we predict physical systems with limited knowledge of its physical properties?* If not, is it possible to transfer hidden physics present in readily accessible visual observations into learning-based physics simulators (Li et al., 2019; Ummenhofer et al., 2020; Sanchez-Gonzalez et al., 2020; Prantl et al., 2022)?

Inspired by human perception, researchers in the field of AI have proposed a series of *intuitive physics* methods (McCloskey et al., 1983; Battaglia et al., 2013; Ehrhardt et al., 2019; Xu et al., 2019; Li et al., 2020) to solve this problem. A typical approach is to build the so-called "*inverse graphics*" models of raw visual observations, which involves training learning-based physical simulators by solving the inverse problem of rendering 3D scenes (Guan et al., 2022; Li et al., 2023). However, NeuroFluid (Guan et al., 2022) needs to finetune the deterministic transition model in response to every physics dynamics associated with new-coming physical properties. PAC-NeRF (Li et al., 2023) adopts heuristic (rather than learnable) simulators and explicitly infers physical properties (such as the viscosity of fluids) given visual observations. It requires an appropriate initial guess of the optimized properties and specifying the fluid type (*e.g.*, Newtonian *vs.* non-Newtonian fluids).

In this paper, we introduce the learning framework of *latent intuitive physics*, which aims to infer hidden fluid dynamics from a 3D video, allowing for the simulation of the observed fluid in novel scenes without the need for its exact physical properties. The framework arises from our intuition that we humans can imagine how a fluid with a specific physical property will move given its initial state by watching a video showcase of it flowing, even though we do not explicitly estimate the exact

---

\*Equal contribution.
†Corresponding author: Yunbo Wang.

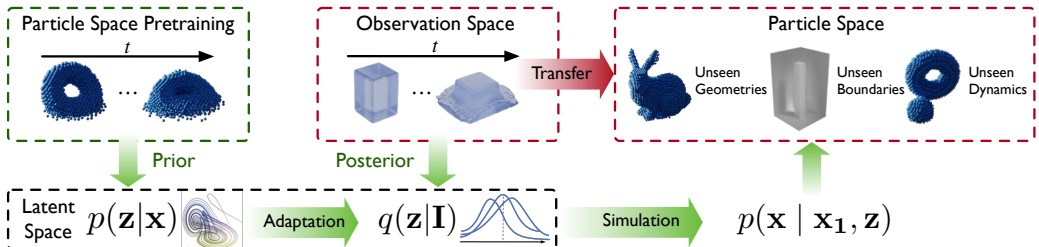

Figure 1: Our approach captures unobservable physical properties from image observations using a parametrized latent space and adapts them for simulating novel scenes with different fluid geometries, boundary conditions, and dynamics. To achieve this, we introduce a variational method that connects the particle space, observation space, and latent space for intuitive physical inference.

values of physical properties. The key idea is to represent the hidden physical properties in visual observations, which may be difficult to observe, using probabilistic latent states $z$ shown in Figure 1. The latent space connects the particle space and visual space to infer and transfer hidden physics with probabilistic modeling. Specifically, our approach includes a probabilistic particle transition module $p(x'|x,z)$[1], a physical prior learner, a particle-based posterior estimator, and a neural renderer, all integrated into a differentiable neural network. The latent features are drawn from trainable marginal distributions $p(z|x)$ that are learned to approximate the visual posterior distribution $q(z|I)$ obtained from a learned neural renderer. By employing probabilistic latent features, our model gains flexibility in modeling complex systems and is capable of capturing uncertainty in the data that a deterministic model may not be able to handle. Once $p(z|x)$ is converged, we embed the prior learner in our probabilistic fluid simulator that is pretrained in particle space containing fluids with a wide range of physical properties. In this way, we transfer the hidden physics from visual observations to particle space to enable novel simulations of unseen fluid geometries, boundary conditions, and dynamics. In our experiments, we demonstrate the effectiveness of *latent intuitive physics* by comparing it to strong baselines of fluid simulation approaches in novel scene simulation, future prediction of the observed dynamics, and supervised particle simulation tasks.

The contributions of this paper can be summarized as follows:

- We introduce *latent intuitive physics*, a learning-based approach for fluid simulation, which infers the hidden properties of fluids from 3D exemplars and transfers this knowledge to a fluid simulator.
- We propose the first *probabilistic particle-based fluid simulation network*, which outperforms prior works in particle-based simulation with varying physical properties.

## 2 RELATED WORK

**Learning-based simulation of particle dynamics.** Recent research has introduced deep learning-based methods to accelerate the forward simulation of complex particle systems and address inverse problems. These methods have shown success in simulating rigid bodies (Battaglia et al., 2016; Han et al., 2022), fluids (Belbute-Peres et al., 2020; Sanchez-Gonzalez et al., 2020; Shao et al., 2022; Prantl et al., 2022), and deformable objects (Mrowca et al., 2018; Li et al., 2019; Sanchez-Gonzalez et al., 2020; Lin et al., 2022). In the context of fluid simulation, DPI-Net (Li et al., 2019) proposes dynamic graphs with multi-step spatial propagation, GNS (Sanchez-Gonzalez et al., 2020) uses message-passing networks, and TIE (Shao et al., 2022) uses a Transformer-based model to capture the spatiotemporal correlations within the particle system. Another line of work includes methods like CConv (Ummenhofer et al., 2020) and DMCF (Prantl et al., 2022), which introduces higher-dimensional continuous convolution operators to model interactions between particles. Different from the approaches for other simulation scenarios for rigid and deformable objects, these models do not assume strong geometric priors such as object-centric transformations or pre-defined mesh topologies (Pfaff et al., 2021; Allen et al., 2022). However, these models are deterministic, assuming that all physical properties are measurable and focus on learning fluid dynamics under known physical properties. Moreover, the stochastic components that commonly exist in the real physical world are not taken into account by the deterministic models.

**Intuitive physics learning with neural networks.** Researchers have explored intuitive physics methods from a range of perspectives. These include heuristic models (Gilden & Proffitt, 1994;

---

[1]Here, we use $x$ to indicate the historical states and $x'$ to indicate the future states in the physical process.

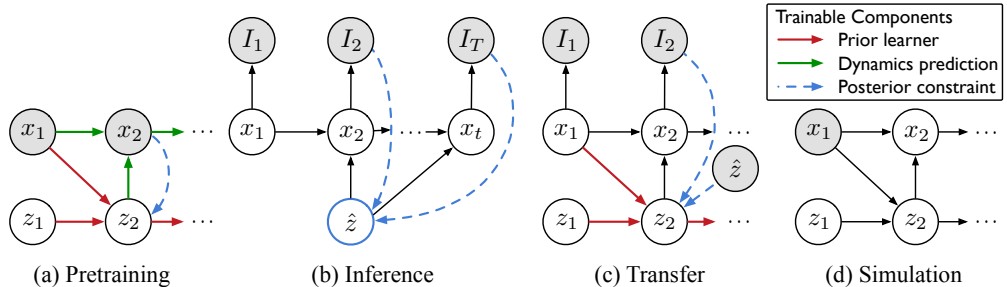

Figure 2: Graphical model of the *pretraining–inference–transfer* pipeline of latent intuitive physics. (a) Particle-space pretraining for probabilistic fluid simulation. (b) Visual posterior optimization from visual observations with a photometric loss. (c) Adaptation of the prior learner to the converged visual posteriors $\hat{z}$. (d) Novel scene simulation with the adapted prior learner. The training parts are highlighted in color. We present details of these training stages in Figure 7 in the appendix.

Sanborn et al., 2013), probabilistic mental simulation models (Hegarty, 2004; Bates et al., 2015), and the cognitive intuitive physics models (Battaglia et al., 2013; Ullman et al., 2017). Recent advances in deep learning typically investigate intuitive physics from different aspects. Some approaches adopt 3D videos to to downstream tasks, such as predicting multi-object dynamics (Driess et al., 2023), fluid dynamic (Guan et al., 2022; Li et al., 2023), system identification (Li et al., 2023), manipulation (Simeonov et al., 2022; Li et al., 2022) or reasoning physics parameters (Li et al., 2020; Chen et al., 2022; Le Cleac'h et al., 2023). However, most works focus on rigid body dynamics (Driess et al., 2023; Li et al., 2020; Le Cleac'h et al., 2023). The most relevant work to our method is NeuroFluid (Guan et al., 2022) and PAC-NeRF (Li et al., 2023), both focusing on fluid dynamics modeling with visual observation. NeuroFluid directly adopts the learning-based simulator from CConv (Ummenhofer et al., 2020). Unlike our approach, it is deterministic and cannot handle the stochastic components or inaccessible physical properties in complex physical scenarios. PAC-NeRF employs specific non-learnable physics simulators tailored to different types of fluids. When using PAC-NeRF to solve inverse problems, users need to make an appropriate initialization for the optimized parameters based on the category of fluid observed.

## 3 PROBLEM FORMULATION

We study the inverse problem of fluid simulation, which refers to learning inaccessible physical properties by leveraging visual observations. Specifically, we consider a dynamic system where we only have a single sequence of observations represented as multi-view videos $\{I_t^m\}_{t=1:T}^{m=1:M}$, where $I_t^m$ represents a visual observation received at time $t$ from view $m$. We want to predict the future states of the system when it appears in novel scenes. Let $\mathbf{x}_t = (x_t^1, \ldots, x_t^N) \in \mathcal{X}$ be a state of the system at time $t$, where $x_t^i = (p_t^i, v_t^i)$ represents the state of the $i^{\text{th}}$ particle that involves the position $p_t^i$ and velocity $v_t^i$, with $v_t^i$ being the time derivative of $p_t^i$. The dynamics of particles $\{\mathbf{x}_1, \ldots, \mathbf{x}_T\}$ is jointly governed by a set of physical properties, such as density, viscosity, and pressure. These properties are hidden and need to be inferred in visual observations. To bridge the gap between particle simulators and the visual world with a varying set of physical properties in a unified framework, we introduce a set of latent features $\mathbf{z}_t = (z_t^1, \ldots, z_t^N)$, where $z_t^i$ is the latent feature attached to each particle. As shown in Figure 2(a), the particle state transition function can thus be represented as $\mathbf{x}_t \sim p(\mathbf{x}_{t-1}, \mathbf{z}_t)$, where $\mathbf{z}_t \sim p(\mathbf{x}_{1:t-1}, \mathbf{z}_{t-1})$. As the explicit physical properties are inaccessible, we can infer latent distribution $p(\mathbf{z}_t \mid \mathbf{x}_{1:t})$ from $q(\mathbf{z} \mid I_{1:T})$. The final goal is to simulate novel scenes with new initial and boundary conditions based on learned physics (see Figure 2(d)).

## 4 LATENT INTUITIVE PHYSICS

In this section, we introduce *latent intuitive physics* for fluid simulation, which enables the transfer of hidden fluid properties from visual observations to novel scenes. As shown in Figure 3, our model consists of four network components: the probabilistic particle transition module ($\theta$), the physical prior learner ($\psi$), the particle-based posterior estimator ($\xi$), and the neural renderer ($\phi$). The training pipeline involves three stages shown in Figure 2—*pretraining*, *inference*, and *transfer*:

a) Pretrain the probabilistic fluid simulator on the particle dataset, which involves the particle transition module $p_\theta(\mathbf{x}_t \mid \mathbf{x}_{t-1}, \mathbf{z}_t)$, the prior learner $p_\psi(\tilde{\mathbf{z}}_t \mid \mathbf{x}_{1:t-1}, \tilde{\mathbf{z}}_{t-1})$, and the posterior $q_\xi(\mathbf{z}_t \mid \mathbf{x}_{1:t}, \mathbf{z}_{t-1})$. The prior module reasons about latent features from historical particle states.

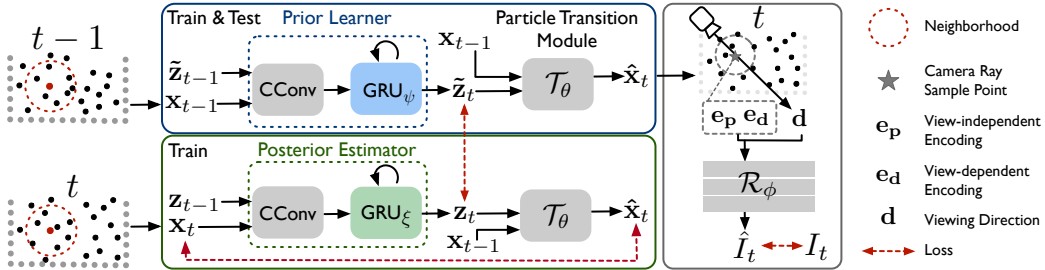

Figure 3: Our model consists of four network components parametrized by $\theta, \psi, \xi, \phi$ respectively. We present the losses for pretraining the simulator and the renderer. For schematics of other training stages (*i.e.*, visual posterior inference and prior adaptation), please refer to Figure 7 in the appendix.

b) Infer the visual posterior latent features $\hat{z}$ from consecutive image observations of a specific fluid, which is achieved by optimizing a differentiable neural renderer ($\phi$).

c) Train the prior learner $p_\psi(\tilde{z}_t \mid x_{1:t-1}, \tilde{z}_{t-1})$ to approximate the converged distribution of $\hat{z}$, which enables the transfer of inaccessible physical properties from the visual world to the simulator.

## 4.1 STAGE A: PROBABILISTIC FLUID SIMULATOR PRETRAINING

Building a probabilistic model has significant advantages for fluid simulation: First, it allows us to predict fluid dynamics without knowing true physical parameters. Instead, it relies on inferring latent features from consecutive inputs. This is valuable because many complex physical phenomena naturally involve stochastic components. Second, when provided with visual observations, we can seamlessly integrate the probabilistic fluid simulation into our variational inference method via the latent space. Next, we introduce how to infer physical properties from particle data.

The architecture of our probabilistic fluid simulator is shown in Figure 3 (Left). Particle states $x_t$ itself is not a feature but simply defines the particle's state in spatial space. We use the continuous convolution (CConv) (Ummenhofer et al., 2020) as a feature encoder to get feature representations of each particle. Inspired by traditional Smoothed Particle Hydrodynamics (SPH) methods, CConv predicts particle features by aggregating its neighbors' features in a continuous and smooth manner. Static particles such as boundaries, are processed similarly but with the particle positions and normal vectors as input (see the literature by Ummenhofer et al. (2020)). Since the invisible physical properties cannot be inferred from a single state, we use a GRU to gather historical information and infer the distribution of the prior latents. The prior learner is trained along with a separate posterior estimator $q_\xi(z_t \mid x_{1:t}, z_{t-1})$ (not used at test time). The models can be written as

$$\text{Prior: } \tilde{z}_t \sim \text{GRU}_\psi(\text{CConv}(\tilde{z}_{t-1}, x_{t-1})); \quad \text{Posterior: } z_t \sim \text{GRU}_\xi(\text{CConv}(z_{t-1}, x_t)), \quad (1)$$

where $z_{t=1}$ and $\tilde{z}_{t=1}$ are zero-initialized. The posterior estimator takes $x_t$ as input, *i.e.*, the target of the prediction. The prior and posterior latents are sampled from distinct Gaussian distributions, with their parameters determined by predicted means and variances by two GRUs. During training, we align their distributions through KL divergence. Notably, our approach assumes time-varying and particle-dependent fluid properties, which aligns with conventional SPH methods (please refer to the work by Bender & Koschier (2015)). This approach empirically achieves better performance than optimizing a global latent variable, as demonstrated in our experiments.

For the particle transition module $p_\theta(x_t \mid x_{t-1}, z_t)$, we adopt another CConv with additional inputs of $z_t$ drawn from the inferred latent distribution. This allows the module to incorporate the previous states $x_{t-1}$ and corresponding latent features for future prediction. As the stochastic physical component has been captured by $z_t$, we employ a deterministic architecture for the particle transition module: $\hat{x}_t \triangleq \{(\hat{p}_t^i, \hat{v}_t^i)\}_{i=1:N} \sim \mathcal{T}_\theta(x_{t-1}, z_t)$. During training, the latent posteriors $z_t$ are used as inputs of $\mathcal{T}_\theta$. At test time, we use the latent priors $\tilde{z}_t$ instead. The fluid simulator is trained with

$$\mathcal{L}_{\theta,\psi,\xi} = \mathbb{E}\Big[\frac{1}{N}\sum_{i=1}^{N} w_i\big\|\hat{p}_t^i - p_t^i\big\|_2^\gamma + \beta\, \mathcal{D}_{KL}\big(q_\xi\left(z_t \mid x_{1:t}, z_{t-1}\right) \| \, p_\psi\left(\tilde{z}_t \mid x_{1:t-1}, \tilde{z}_{t-1}\right)\big)\Big]. \quad (2)$$

Similar to the previous work (Li et al., 2019; Prantl et al., 2022; Ummenhofer et al., 2020; Sanchez-Gonzalez et al., 2020), we use the $\|\cdot\|_2^\gamma$ error between the predicted position and the ground-truth positions, and weight it by the neighbor count to form the reconstruction loss. Specifically, we use $w_i = \exp(-\frac{1}{c}\mathcal{N}(\hat{p}_t^i))$, where $\mathcal{N}(\hat{p}_t^i)$ denotes the number of neighbors for the predicted particle $i$ and $c$ is the average neighbor count.

## 4.2 STAGE B: VISUAL POSTERIOR INFERENCE

Here we introduce how to solve the inverse problem by inferring scene-specific visual posteriors, where the visual observation governs the physical properties. In this stage, the pretrained particle transition module $\mathcal{T}_\theta$ infers the input visual posterior, which facilitates the adaptation of the prior learner in the latent space in Stage C. To this end, the particle transition module is combined with a differentiable neural renderer $\mathcal{R}_\phi$ that provides gradients backpropagated from the photometric error over sequences in observation space. Note that only visual observations are available in the following.

**Neural renderer.** To enable joint modeling of the state-to-state function of fluid dynamics and the state-to-graphics mapping function, we integrate the probabilistic particle transition module with the particle-driven neural renderer (PhysNeRF) in NeuroFluid (Guan et al., 2022) in a differentiable framework. PhysNeRF uses view-independent particle encoding $\mathbf{e_p}$ and view-dependent particle encoding $\mathbf{e_d}$ to estimate the volume density $\sigma$ and the color $\mathbf{c}$ of each sampled point along each ray $\mathbf{r}(t) = \mathbf{o} + t\mathbf{d}$, such that $(\mathbf{c}, \sigma) = \mathcal{R}_\phi(\mathbf{e_p}, \mathbf{e_d}, \mathbf{d})$. In this way, it establishes correlations between the particle distribution and the neural radiance field. Unlike the original PhysNeRF, we exclude the position of the sampled point from the inputs to the rendering network, which enhances the relationships between the fluid particle encodings and the rendering results. The neural renderer $\mathcal{R}_\phi$ is pretrained on multiple visual scenes so that it can respond to various particle-based geometries.

**Initial states estimation.** NeuroFluid assumes known initial particle states. However, when only visual observations are available, estimating the initial particle states $\mathbf{x}_{t=1}$ becomes necessary. These estimated initial states are used to drive the neural renderer ($\mathcal{R}\phi$) for generating visual predictions at the first time step and also to initiate the particle transition module ($\mathcal{T}_\theta$) for simulating subsequent states. We estimate the initial particle positions using the voxel-based neural rendering technique (Liu et al., 2020; Sun et al., 2022; Müller et al., 2022) at the first time step. During training, we maintain an occupancy cache to represent empty *vs.* nonempty space and randomly sample fluid particles within each voxel grid in the visual posterior inference stage (see Appendix D.2.3 for details).

**Optimization.** We first finetune the neural renderer $\mathcal{R}_\phi$ on current visual observation with initial state estimation $\hat{\mathbf{x}}_{t=1}$. Then the parameters of $\mathcal{T}_\theta$ and $\mathcal{R}_\phi$ are frozen and we initialize a set of visual posterior latents $\hat{\mathbf{z}}$, such that $\hat{\mathbf{x}}_t = \mathcal{T}_\theta(\hat{\mathbf{x}}_{t-1}, \hat{\mathbf{z}})$. In practice, we attach a particle-dependent Gaussian distribution $\mathcal{N}(\hat{\mu}^i, \hat{\sigma}^i)$ with trainable parameters to each particle $i$. At each time step, we sample $\hat{z}^i \sim \mathcal{N}(\hat{\mu}^i, \hat{\sigma}^i)$ to form $\hat{\mathbf{z}} = (\hat{z}^1, \ldots, \hat{z}^N)$. The output of the neural renderer $\mathcal{R}_\phi$ is denoted as $\hat{\mathbf{C}}(\mathbf{r}, t)$. We optimize the distributions of the visual posterior latent $\hat{\mathbf{z}}$ over the entire sequence by backpropagating the photometric error $\sum_{\mathbf{r},t} \|\hat{\mathbf{C}}(\mathbf{r}, t) - \mathbf{C}(\mathbf{r}, t)\|$. We summarize the overall training algorithm in the Alg. 1 in the appendix.

## 4.3 STAGE C: PHYSICAL PRIOR ADAPTATION

The visual posteriors learned in the previous stage are specific to the estimated particles within the observed scene and cannot be directly applied to simulate novel scenes. Therefore, in this stage, we aim to adapt the hidden physical properties encoded in the visual posterior to the physical prior learner $p_\psi$. Instead of finetuning the entire particle transition model as NeuroFluid (Guan et al., 2022) does, we only finetune the prior learner module. Due to the unavailability of the ground truth supervision signal in particle space, tuning all parameters in the transition model in visual scenes might lead to overfitting problems, as the transition model may forget the pre-learned knowledge of feasible dynamics, or learn implausible particle transitions even if it can generate similar fluid geometries that are sufficient to minimize the image rendering loss. Specifically, we perform forward modeling on particle states $\hat{\mathbf{x}}_t$ by applying $\hat{\mathbf{x}}_t = \mathcal{T}_\theta(\mathbf{x}_{t-1}, \tilde{\mathbf{z}}_t)$, where $\tilde{\mathbf{z}}_t$ is sampled from the distribution $p_\psi(\tilde{\mathbf{z}}_t \mid \mathbf{x}_{1:t-1}, \tilde{\mathbf{z}}_{t-1})$ predicted by the prior learner. To transfer the visual posterior to $p_\psi$, we finetune the prior learner by minimizing the distance between its generated distribution and the pre-learned visual posteriors $\{\mathcal{N}(\hat{\mu}^i, \hat{\sigma}^i)\}^{i=1:N}$ with $\mathcal{T}_\theta$ and $\mathcal{R}_\phi$ fixed. The volume rendering loss is still used for supervision as well. The entire training objective is

$$\mathcal{L}_\psi = \sum_{\mathbf{r},t} \|\hat{\mathbf{C}}(\mathbf{r}, t) - \mathbf{C}(\mathbf{r}, t)\| + \beta\, \mathcal{D}_{KL}\left(q(\hat{\mathbf{z}}) \,\|\, p_\psi\left(\tilde{\mathbf{z}}_t \mid \mathbf{x}_{1:t-1}, \tilde{\mathbf{z}}_{t-1}\right)\right). \tag{3}$$

With the finetuned physical prior learner $p_\psi$, we can embed it into the probabilistic fluid simulator to perform novel simulations on unseen fluid geometries, boundary conditions, and dynamics with identical physical properties, which brings the simulation back to the particle space.

Table 1: Quantitative results of average prediction error $\bar{d}$ on unseen fluid geometries and boundary conditions given $\mathbf{x}_{t=1}$. We present the mean and standard deviation of 10 independent samples drawn from our model (for *Ours* only). *Geometry* means unseen fluid initial positions while *Boundary* means unseen boundary conditions. Each physical property set is trained with a single visual observation.

| METHOD | $\rho = 2000, \nu = 0.065$ | | $\rho = 1000, \nu = 0.08$ | | $\rho = 500, \nu = 0.2$ | |
| --- | --- | --- | --- | --- | --- | --- |
| | GEOMETRY | BOUNDARY | GEOMETRY | BOUNDARY | GEOMETRY | BOUNDARY |
| CCONV | 52.49 | 64.29 | 51.40 | 56.33 | 40.67 | 53.28 |
| NEUROFLUID | 65.01 | 73.55 | 59.79 | 60.46 | 40.88 | 50.73 |
| SYS-ID | 156.59 | 179.11 | 127.06 | 140.27 | 58.71 | 71.80 |
| PAC-NERF | 51.10 | 59.61 | 51.33 | 56.84 | 40.97 | 62.05 |
| OURS | **34.54±0.55** | **39.86±0.90** | **33.11±0.50** | **37.79±0.84** | **39.03±0.79** | **47.25±1.26** |

Table 2: Quantitative results of the average future prediction error $\bar{d}$ on observed scenes. We compute the mean results and standard deviations of 10 independent samples drawn from our model. Notably, NeuroFluid learns to overfit the observed scene by tuning the entire particle transition simulator.

| METHOD | $\rho = 2000, \nu = 0.065$ | $\rho = 1000, \nu = 0.08$ | $\rho = 500, \nu = 0.2$ |
| --- | --- | --- | --- |
| CCONV | 107.27 | 103.95 | 54.37 |
| NEUROFLUID | 85.45 | 73.04 | **33.22** |
| SYS-ID | 37.25 | 36.28 | 42.71 |
| PAC-NERF | 44.42 | 42.82 | 50.05 |
| OURS | **32.41±0.17** | **32.97±0.71** | 41.15±0.71 |

## 5 EXPERIMENTS

### 5.1 EVALUATION OF VISUAL PHYSICAL INFERENCE

**Settings.** To evaluate *latent intuitive physics* for inferring and transferring unknown fluid dynamics from visual observations, we generate a *single* sequence using *Cuboid* as the fluid body (not seen during pretraining in Stage A). The fluid freely falls in a default container. The sequence contains 60 time steps, where the first 50 time steps are used for visual inference, while the last 10 time steps are reserved for additional validation on future rollouts. We use Blender (Community, 2018) to generate multi-view images with 20 randomly sampled viewpoints. We assess the performance of our fluid simulator after transferring hidden properties from visual scenes through: (1) simulating novel scenes with fluid geometries that were not seen during training (*Stanford Bunny*, *Sphere*, *Dam Break*), (2) simulating novel scenes with fluid dynamics with previously unseen boundaries. We conduct experiments across three distinct sets of physical properties. In each evaluation set, we impose random rotation, scaling, and velocities on fluid bodies. Please refer to Appendix C & D.2 for more information on visual examples, novel scenes for evaluation, and implementation details. Following Ummenhofer et al. (2020), we compute the average Euclidean distance from the ground truth particles to the closest predicted particles as the evaluation metric, where the average prediction error is $\bar{d} = \frac{1}{T \times N} \sum_t \sum_i \min_{\hat{p}_t} ||p_t^i - \hat{p}_t||_2$. Please see Appendix D.1 for details.

**Compared methods.** We use four baseline models. *CConv* (Ummenhofer et al., 2020) learns particle dynamics without physical parameter inputs. *NeuroFluid* (Guan et al., 2022) grounds fluid dynamics in observed scenes with a particle-driven neural render. *PAC-NeRF* (Li et al., 2023) and *System Identification (Sys-ID)* estimate explicitly physical parameters in the observed scenes. PAC-NeRF employs an MPM simulator. Sys-ID utilizes a CConv simulator that takes learnable physical parameters as inputs. It also employs the same neural renderer as our approach. All models are trained on *Cuboid* and tested on novel scenes. Please refer to Appendix D.2.1 for more details.

**Novel scene simulation results.** We evaluate the simulation results of our approach given only the true initial particle states $\mathbf{x}_{t=1}$ of the novel scenes. Table 1 shows the average prediction error across all testing sequences. We predict each sequence 10 times with different $\mathbf{z}_t$ drawn from the same prior learner and calculate the standard deviation of the errors. We can see that the adapted probabilistic fluid simulator from the visual posteriors significantly outperforms the baselines on novel scenes across all physical properties. Though our model is trained on scenes with the default fluid boundary, it shows the ability to generalize to unseen boundary conditions. Figure 4 showcases the qualitative results. Among the baselines, Sys-ID underperforms in novel scene simulation, as it requires accurate

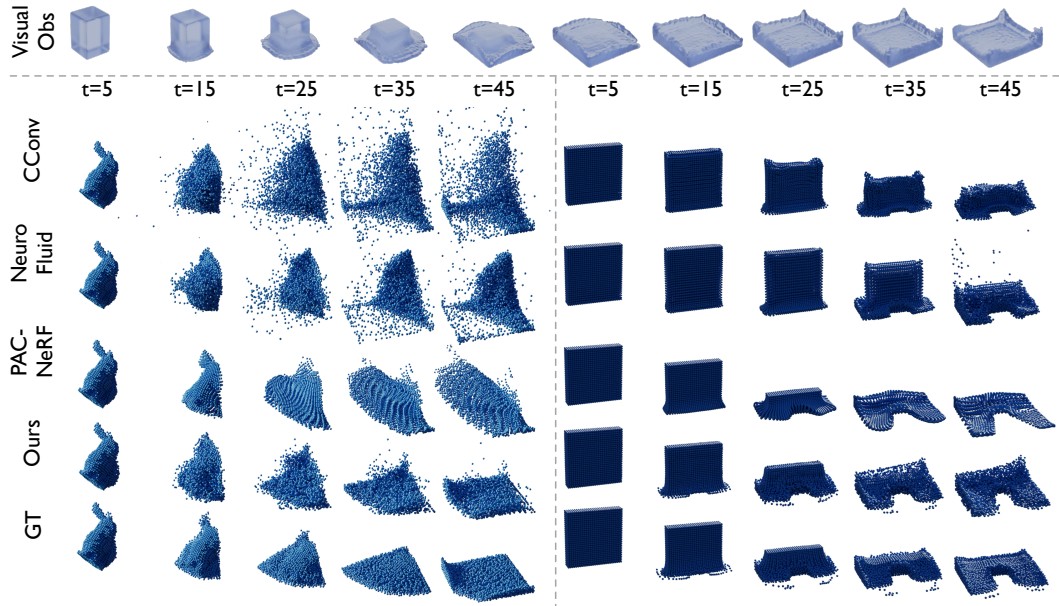

Figure 4: The first row shows the visual observation on the observed scene with physical parameters $\rho = 2000, \nu = 0.065$. Rows 2-6 show qualitative results of simulated particles on novel scenes (Left: unseen geometries, Right: unseen boundaries). More qualitative results are provided in the appendix.

parameter inference from visual observations. In contrast, our model encodes the hidden properties with higher-dimensional latent variables, providing a stronger feature representation for the unknown properties in a physical system. PAC-NeRF employs an MPM simulator, which inherently produces more accurate and stable simulation results than the learning-based simulators used by other models trained on limited data. Despite this advantage, PAC-NeRF tends to overfit the observed scenes and yields degraded performance when applied to novel scenes. More results are shown in Appendix F.

**Future prediction results of the observed scenes.** We predict the particle dynamics of the observed scene *Cuboid* for 10 time steps into the future. As shown in Table 2, our model performs best in most cases. NeuroFluid slightly outperforms our model on $\rho = 500, \nu = 0.2$. Since NeuroFluid jointly optimizes the entire transition model and renderer on the observed scene, it is possible to overfit the observed scene and produce plausible future prediction results. However, it fails to generalize to novel scenes as shown in Table 1. Unlike NeuroFluid, our approach adapts the physical prior learner to visual scenes, without training a probabilistic physics engine. By leveraging knowledge from the pretraining stage, the transition model is less prone to overfitting on the observed scene. This not only enhances the generalization ability but also significantly reduces the training burden.

## 5.2 EVALUATION OF PROBABILISTIC FLUID SIMULATOR

To validate whether our approach can infer hidden physics from particle data in latent space, we evaluate the pretrained probabilistic fluid simulator $(\theta, \psi, \xi)$ in a particle dataset generated with DFSPH (Bender & Koschier, 2015), which simulates fluids with various physical parameters (*e.g.*, viscosity $\nu$, density $\rho$) falling in a cubic box. Each scene contains 273-19,682 fluid particles and 200 time steps. To assess the simulation performance under incomplete measurement of physical parameters, the true parameters are invisible to simulators. See the Appendix C.1 for more details.

We compare our probabilistic fluid simulator with four representative particle simulation approaches, based on graph neural networks, continuous convolution models, and Transformer, *i.e.*, DPI-Net (Li et al., 2019), CConv (Ummenhofer et al., 2020), DMCF (Prantl et al., 2022), and TIE (Shao et al., 2022). Following Ummenhofer et al. (2020), given two consecutive input states $\mathbf{x}_{t-1:t}$, we compute the errors of the predicted particle positions w.r.t. the true particles: $d_{t+\tau} = \frac{1}{N} \sum_i ||p_{t+\tau}^i - \hat{p}_{t+\tau}^i||_2$, for the next two steps ($\tau \in \{1, 2\}$). To assess the long-term prediction ability, we also calculate the average distance $\bar{d}$ from true particle positions to the predicted particles over the entire sequence. The first 10 states are given, and the models predict the following 190 states. From Table 3, our model performs best in both short-term and long-term prediction. The qualitative result of long-term prediction is shown in Figure 16 in Appendix F. These results showcase that our probabilistic fluid simulation method provides an effective avenue for intuitive physics learning.

Table 3: Quantitative comparisons of fluid simulators on the particle dataset with inaccessible physical properties. We present the prediction errors for the next two frames $d_{t+1}$ and $d_{t+2}$, and the averaged prediction error $\bar{d}$ over the whole sequence for 190 prediction time steps.

| METHODS | $d_{t+1}$ | $d_{t+2}$ | $\bar{d}$ |
|---|---|---|---|
| DPI-NET (LI ET AL., 2019) | 0.95 | 2.99 | 90.21 |
| CCONV (UMMENHOFER ET AL., 2020) | 0.34 | 1.03 | 44.79 |
| DMCF (PRANTL ET AL., 2022) | 0.54 | 1.23 | 39.70 |
| TIE (SHAO ET AL., 2022) | 0.52 | 1.36 | 41.82 |
| OURS | **0.31±0.003** | **0.94±0.011** | **38.37±0.860** |

Table 4: Results of $\bar{d}$ on generalization to unseen fluid dynamics given $\mathbf{x}_{t=1}$. *Global Latent* is a variant of our model with a scene-specific global latent.

| METHODS | CCONV | NEUROFLUID | GLOBAL LATENT | OURS |
|---|---|---|---|---|
| OBSERVED | 56.92 | 54.67 | 60.63 | **36.03 ± 0.80** |
| UNSEEN | 46.83 | 54.50 | 90.51 | **44.25 ± 1.36** |

## 5.3 GENERALIZATION TO DYNAMICS DISCREPANCIES

To validate the generalization ability of our approach across the discrepancies in fluid dynamics patterns, we consider a more complex scenario that contains a mixture of two different fluids. Specifically, we use the pretrained probabilistic fluid simulator discussed in Sec 5.2, and adapt the model to a visual scene containing two heterogeneous fluids interacting with each other. We use two prior learners with the same initialization and a single particle transition module to learn the hidden physics of different fluid drops separately. Table 4 and Figure 5 present both quantitative and qualitative results. Our approach showcases robust generalization ability when dealing with visual scenes containing fluid dynamics that diverge significantly from the patterns in the pretrained dataset.

Furthermore, we explore the performance of an alternative method that employs scene-specific, time-invariant latent variables (*i.e.*, *Global Latent* in Table 4 and Figure 5). From these results, we find that optimizing time-varying latent features individually for each particle is more effective than optimizing two sets of global latents, each designated for a particular fluid type.

## 5.4 ABLATION STUDY

To verify the effectiveness of transferring the learned visual posterior $\hat{\mathbf{z}}$ to the physical prior learner, we experiment with different variants of our method. The result is shown in Table 5. *w/o StageB*

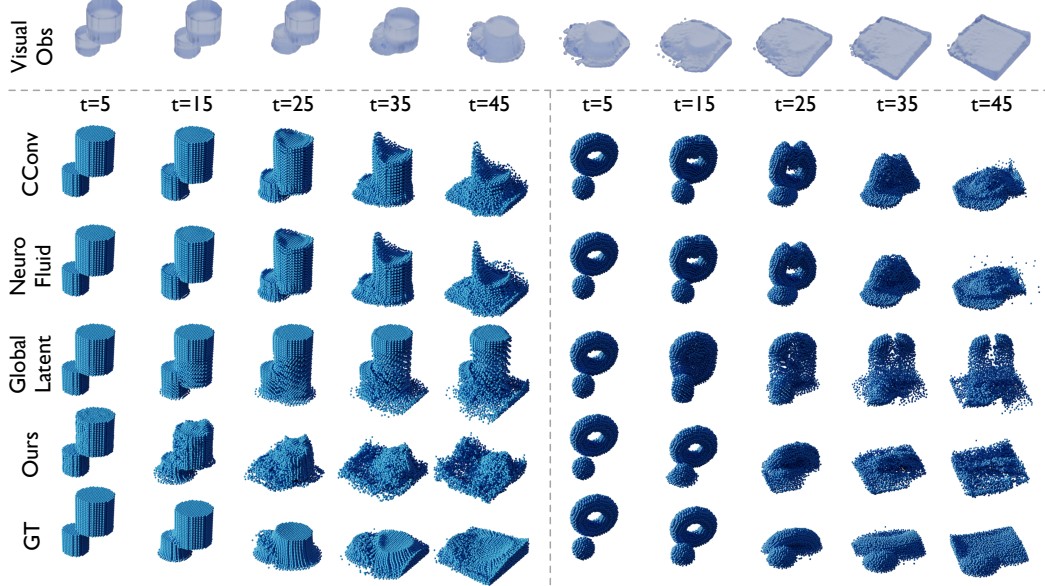

Figure 5: Qualitative results on generalization to unseen dynamics of heterogeneous fluids. We present simulation results on the observed scene (Left) and a novel scene (Right).

Table 5: Ablation study for each training stage in our pipeline. We report mean prediction error ($\bar{d}$) resulting from the removal of Stage B, Stage C, or both, based on 10 samples from our model. The future prediction error is reported on observed scenes and the prediction errors on unseen fluid geometries given $\mathbf{x}_{t=1}$. **Ours**[†]**:** Using true initial states in the observed scenes for visual inference.

| | $\rho = 2000, \nu = 0.065$ | | $\rho = 1000, \nu = 0.08$ | | $\rho = 500, \nu = 0.2$ | |
|---|---|---|---|---|---|---|
| METHODS | OBSERVED | UNSEEN | OBSERVED | UNSEEN | OBSERVED | UNSEEN |
| W/O STAGE C | 33.22 | N/A | 32.72 | N/A | **37.04** | N/A |
| W/O STAGE B | 37.55 | 42.43 | 35.75 | 46.98 | 41.71 | 41.40 |
| OURS | 32.41 | 34.54 | 32.97 | **33.11** | 41.15 | 39.03 |
| OURS[†] | **28.99** | **33.34** | **31.87** | 38.84 | 39.70 | **38.41** |

refers to the model without transferring posterior latent distribution $\hat{\mathbf{z}}$ to the physical prior learner and directly finetunes it on visual observations. *w/o Stage C* refers to without adapting the physical prior learner. In this case, since we only have features $\hat{\mathbf{z}}$ attached to each particle, we cannot simulate novel scenes of various particle numbers. The results show that the posterior latent distribution can make training more stable by restricting the range of distribution in latent space, and the transfer learning of the prior learner enables the prediction in novel scenes. In addition, we further investigate the performance gap between feeding the ground truth initial state and the estimated initial state to the probabilistic fluid simulator. We find that models utilizing estimated initial states yield comparable performance results when compared to those using ground truth initial states.

## 6 POSSIBILITIES OF REAL-WORLD EXPERIMENTS

The primary focus of this paper is to explore the feasibility of a new learning scheme for intuitive physics. Using synthetic data can greatly facilitate the evaluation of our model, as the simulation results can be directly quantified using particle states; whereas real-world scenes would necessitate more advanced fluid flow measurement techniques like *particle image velocimetry*. For a similar reason, earlier attempts like NeuroFluid and PAC-NeRF are also evaluated on synthetic data. Nevertheless, we acknowledge that real-world validation is meaningful and challenging, and so make our best efforts to explore the possibility of implementing *latent intuitive physics* in real-world scenarios. As shown in Figure 6, we capture RGB images of dyed water in a fluid tank at a resolution of $1,200 \times 900$. To cope with the complex and noisy visual environment, we adopt NeRFREN (Guo et al., 2022) to remove the

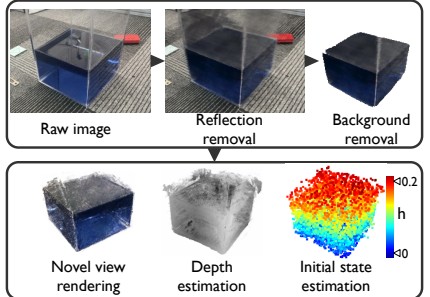

Figure 6: Our pipeline and intermediate results for real-world experiments. We capture images of dyed water in a fluid tank and estimate the initial states. We will explore dynamic scenes in future work.

reflection and refraction and segment the fluid body by SAM (Kirillov et al., 2023). The preprocessed images are then used to estimate fluid positions using our proposed initial state estimation module. We carefully discuss the pipeline in Appendix G and include a video in the supplementary. Another notable challenge of the complete real-world experiments is to acquire high frame-rate images with synchronized cameras across multiple viewpoints. We leave this part for future work.

## 7 CONCLUSION

In this paper, we presented latent intuitive physics, which learns the hidden physical properties of fluids from a 3D video. The key contributions include: 1) a probabilistic fluid simulator that considers the stochastic nature of complex physical processes, and 2) a variational inference learning method that can transfer the posteriors of the hidden parameters from visual observations to the fluid simulator. Accordingly, we proposed the pretraining-inference-transfer optimization scheme for the model, which allows for easy transfer of visual-world fluid properties to novel scene simulation with various initial states and boundary conditions. We evaluated our model on synthetic datasets (similar to existing literature) and discussed its potential in real-world experiments.

## ACKNOWLEDGMENTS

This work was supported by the National Natural Science Foundation of China (Grant No. 62250062, 62106144), the Shanghai Municipal Science and Technology Major Project (Grant No. 2021SHZDZX0102), the Fundamental Research Funds for the Central Universities, the Shanghai Sailing Program (Grant No. 21Z510202133), and the CCF-Tencent Rhino-Bird Open Research Fund.

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

# A MODEL DETAILS

## A.1 OVERALL FRAMEWORK

Figure 7 demonstrates the main architecture of the proposed framework, and Table 6 presents the summary of all model components, including formulation, input, output, training stage, and objective. The entire model includes a probabilistic particle transition module ($\theta$), a physical prior learner ($\psi$), a particle-based posterior module ($\xi$), and a neural renderer ($\phi$). The physical prior learner $p_\psi(\tilde{\mathbf{z}}_t \mid \mathbf{x}_{1:t-1}, \tilde{\mathbf{z}}_{t-1})$ infers latent features $\tilde{\mathbf{z}}_t$ from consecutive particle data, which the transition module $p_\theta(\mathbf{x}_t \mid \mathbf{x}_{t-1}, \mathbf{z}_t)$ can use to predict the next state $\mathbf{x}_t$. In the pre-training stage, a particle-based posterior $q_\xi(\mathbf{z}_t \mid \mathbf{x}_{1:t}, \mathbf{z}_{t-1})$ generates posterior $\mathbf{z}_t$ with the prediction target $\mathbf{x}_t$ as input. It guides the prior learner to generate meaningful latent features that capture hidden properties essential for accurate predictions. During visual inference and transfer in Stage B and C, the neural renderer $\mathcal{R}_\phi$ generates images according to particle positions and viewing directions, which enables photometric error to be backpropagated through the whole differentiable neural network. In the visual posterior inference stage (Stage B), the visual posteriors $\hat{\mathbf{z}}$ are optimized by backpropagating image rendering errors. They serve as adaptation targets for the prior learner, facilitating efficient adaptation to the specific observed fluid.

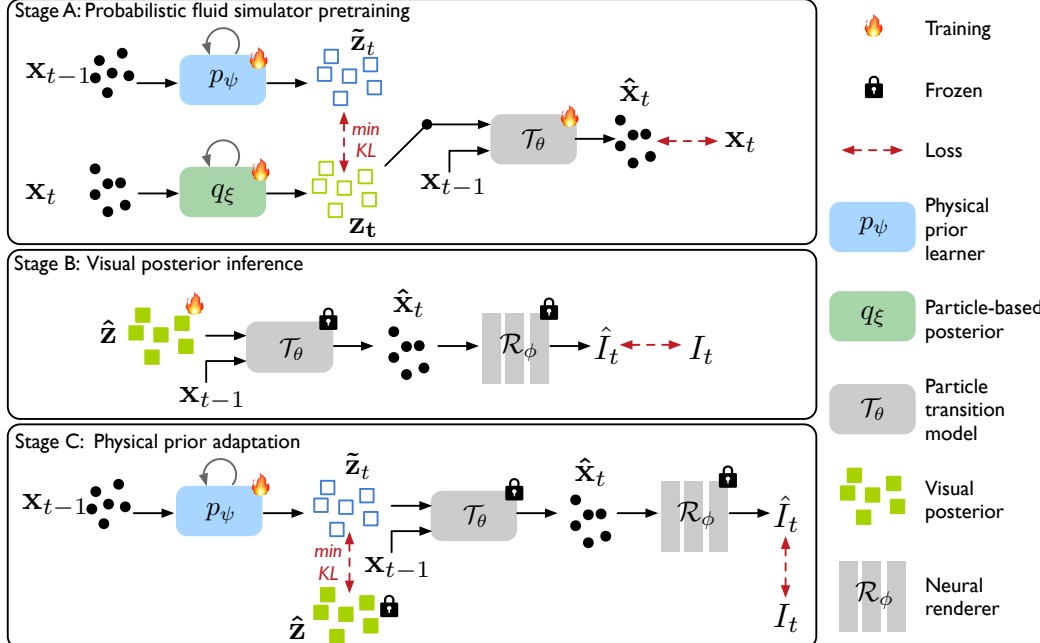

Figure 7: The overview of the proposed framework. The prior learner infers latent features from the data with the help of particle/visual posterior. The latent features are then used in the particle transition module to predict the next state. The neural renderer bridges the gap from the particle space to the visual observation space.

## A.2 PROBABILISTIC FLUID SIMULATOR

The probabilistic fluid simulator is used to infer hidden physics and simulate various kinds of fluid dynamics. Specifically, as shown in Figure 3, our probabilistic fluid simulator consists of three modules: probabilistic particle transition module ($\theta$), physical prior learner ($\psi$) and particle-based posterior module ($\xi$).

We use continuous convolution (CConv) (Ummenhofer et al., 2020) as the shared feature encoder for the prior learner ($\psi$) and the particle-based posterior module ($\xi$) to extract features from particle states. The CConv module predicts particle features in a smooth and continuous way: $(f * g)(x) =$

Table 6: The overview of model components in our method.

| | | |
|---|---|---|
| Particle transition module | Formulation | $p_\theta(\mathbf{x}_t \mid \mathbf{x}_{t-1}, \mathbf{z}_t)$ |
| | Input | Particle state $\mathbf{x}_{t-1}$ and latent feature $\mathbf{z}_t$ |
| | Output | Next particle state $\hat{\mathbf{x}}_t$ |
| | Training stage | Stage A |
| | Objective | Predict the next state according to particle states and latent features |
| Physical prior learner | Formulation | $\tilde{\mathbf{z}}_t \sim p_\psi(\tilde{\mathbf{z}}_t \mid \mathbf{x}_{1:t-1}, \tilde{\mathbf{z}}_{t-1})$ |
| | Input | Particle states $\mathbf{x}_{1:t-1}$ and latent feature $\tilde{\mathbf{z}}_{t-1}$ |
| | Output | Prior latent feature $\tilde{\mathbf{z}}_t$ |
| | Training stage | Stage A & C |
| | Objective | Narrow the gap with the posterior distribution |
| Particle-based posterior | Formulation | $\mathbf{z}_t \sim q_\xi(\mathbf{z}_t \mid \mathbf{x}_{1:t}, \mathbf{z}_{t-1})$ |
| | Input | Future particle state $\mathbf{x}_t$, and previous states and features $(\mathbf{x}_{1:t-1}, \mathbf{z}_{t-1})$ |
| | Output | Posterior latent feature $\mathbf{z}_t$ |
| | Training stage | Stage A |
| | Objective | Facilitate prior learner training when particle states are available |
| Visual posterior | Formulation | $\hat{\mathbf{z}} \sim q(\hat{\mathbf{z}} \mid I_{1:T})$ |
| | Input | N/A |
| | Output | Particle-dependent feature |
| | Training stage | Stage B |
| | Objective | Provide adaptation targets for the prior learner in subsequent Stage C |
| Neural renderer | Formulation | $\mathcal{R}_\phi(\mathbf{e_p}, \mathbf{e_d}, \mathbf{d})$ |
| | Input | Encoding based on predicted particle positions and viewing directions |
| | Output | Rendered image |
| | Training stage | Pretrained on multiple visual scenes |
| | Objective | Backpropagate the photometric error |

$\frac{1}{n(x)} \sum_{i \in \mathcal{N}(x,R)} a\left(x^i, x\right) f_i g\left(\Lambda\left(x^i - x\right)\right)$, with $f$ the input feature function and $g$ as filter function. The input consists of particle positions and the corresponding feature. The normalization $\frac{1}{n(x)}$ can be turned on with the normalization parameter. The per neighbor value $a(x^i, x)$ is a window function to produce a smooth response of the convolution under varying particle neighborhoods. It determines the $i^{\text{th}}$ particle feature by aggregating particle features given its neighbors' features. Thereby, the features of the neighbors are weighted with a kernel function depending on their relative position. The kernel functions themselves are discretized via a regular grid with spherical mapping and contain the learnable parameters. To cope with boundary interactions, we follow the implementation of (Ummenhofer et al., 2020). Specifically, the feature encoder is realized with two separate CConv layers: One is to encode the fluid particles in the neighborhood of each particle location; The other one is to handle the virtual boundary particles in the same neighborhood. The features extracted from these processes are then concatenated to form the inputs for subsequent layers. The following GRUs can summarize the historical information and provide the inferred distribution of latent features about the system. In practice, we use the mean of distributions at the last time step as input of the encoder CConv to avoid excessively noisy inputs.

### A.3 NEURAL RENDERER

We present detailed model architecture of PhysNeRF Guan et al. (2022). As shown in Figure 8, the network is based on fully-connected layers and similar to NeRF (Mildenhall et al., 2020). Unlike original NeRF, PhysNeRF performs volume rendering according to the geometric distribution of neighboring physical particles along a given camera ray. For a sampled point in a ray, a ball query is conducted to identify neighboring fluid particles around the sampled point. These neighboring particles are then parameterized to obtain view-independent and view-dependent encodings $\mathbf{e_p}, \mathbf{e_d}$, which are used in the volume rendering subsequently. A multilayer perception (MLP) is trained to map these encodings along with view direction $\mathbf{d}$ to volume density $\sigma$ and emitted color $\mathbf{c}$ of each sampled point along each ray $\mathbf{r}(t) = \mathbf{o} + t\mathbf{d}$, such that $(\mathbf{c}, \sigma) = \mathcal{R}_\phi(\mathbf{e_p}, \mathbf{e_d}, \mathbf{d})$. Different from original PhysNeRF, we exclude the position of the sampled point from the inputs to the MLP network, which enhances the relationships between the fluid particle encodings and the rendering results. The

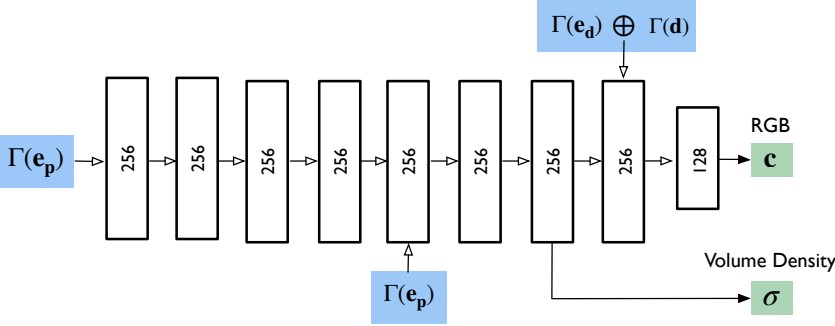

Figure 8: Network architecture of PhysNeRF. The inputs contain view-independent particle encoding $\mathbf{e_p}$, view-dependent particle encoding $\mathbf{e_d}$ and camera viewing direction $\mathbf{d}$. The outputs are volume density $\sigma$ and RGB color $\mathbf{c}$.

view-independent encodings $\mathbf{e_p}$ includes 3 parts: fictitious particle center $\mathbf{p}_c$, soft particle density $\sigma_\mathrm{p}$ and radial deformation $\boldsymbol{v}_\mathrm{D}$, calculated as:

$$\text{Fictitious particle center:} \quad p_\mathrm{c} = \frac{1}{K} \sum_{p^i \in \mathcal{N}(\mathbf{r}(t), r_s)} w_i p^i$$

$$\text{Soft particle density:} \quad \sigma_\mathrm{p} = \sum_i w_i \tag{4}$$

$$\text{Radial deformation:} \quad \boldsymbol{v}_\mathrm{D} = \frac{1}{K} \sum_i \left\| \left\| p^i - \mathbf{r}(t) \right\| - \frac{1}{K} \sum_i \left\| p^i - \mathbf{r}(t) \right\| \right\|_2,$$

where $\mathcal{N}(\mathbf{r}(t), r_s)$ is the ball query neighborhood within radius $r_s$ of the sampled camera ray point $\mathbf{r}(t)$ and $w_i = \max(1 - (\frac{\|p^i - \mathbf{r}(t)\|_2}{r_s})^3, 0)$. The view-dependent encoding $\mathbf{e_d}$ includes the normalized view direction to the fictitious particle center, which is an important reference direction for the network to infer the refraction and reflection, calculated as:

$$\boldsymbol{d}_\mathrm{c} = (p_\mathrm{c} - \mathbf{o}) \,/\, \|p_\mathrm{c} - \mathbf{o}\|_2 \,. \tag{5}$$

Finally, we take into account all the above physical quantities and derive the view-independent encoding and the view-dependent encoding as

$$\mathbf{e_p} = (p_\mathrm{c}, \sigma_\mathrm{p}, \boldsymbol{v}_\mathrm{D}), \quad \mathbf{e_d} = \boldsymbol{d}_c. \tag{6}$$

Following (Mildenhall et al., 2020; Liu et al., 2020; Sun et al., 2022), we apply positional encoding $\Gamma(\cdot)$ to every input. In our experiments, we set the maximum encoded frequency $L = 10$ for $\Gamma(\mathbf{e_p})$ and $L = 4$ for $\Gamma(\mathbf{e_d}), \Gamma(\mathbf{d})$. We optimize PhysNeRF in a coarse-to-fine manner (Mildenhall et al., 2020). For the fine MLP network, the search radius of particle encoding of $\mathbf{e_p}$ and $\mathbf{e_d}$ is set as 3 times the particle radius and we consider 20 fluid particles within this search radius. For the coarse MLP network, the search radius scale and the number of encoding neighbors are set as 1.3 times the parameters of the fine network.

## A.4 HYPERPARAMETERS

Table 7 shows the hyperparameters used in experiments.

## B TRAINING ALGORITHM

Algorithm 1 gives detailed descriptions of the computation flow of the training process.

Table 7: Hyperparameters of the probabilistic fluid simulator and neural renderer

| Name | Symbol | Value |
|---|---|---|
| Stage A | | |
| KL loss scale | $\beta$ | 0.1 |
| Distance $\|\cdot\|_2^\gamma$ | $\gamma$ | 0.5 |
| Gaussian latent dimension | — | 8 |
| Number of CConv blocks of shared feature encoder | — | 3 |
| Number of CConv blocks of Particle dynamics module | — | 4 |
| Number of GRU layers | — | 1 |
| Neighborhood radius | — | $4.5 \times$ particle radius |
| Stage B | | |
| Image size | H×W | $400 \times 400$ |
| Coarse sample points | $N_c$ | 64 |
| Fine sample points | $N_f$ | 128 |
| Particle encoding radius of $\mathbf{e_p}$ and $\mathbf{e_d}$ | $r_s$ | $3 \times$ particle radius |
| Particle encoding neighbors | $nn$ | 20 |
| Coarse encoding scale | $s$ | 1.3 |
| Stage C | | |
| KL loss scale | $\beta$ | 0.01 |

---

**Algorithm 1:** Learning procedures in the visual scene

---

1   **Given:** Multi-view observation $\{I_t^{1:m}\}$, pretrained $\mathcal{T}_\theta$, prior learner $p_\psi$ (Stage A) and $\mathcal{R}_\phi$.

2   **Unknown:** Particle state $\mathbf{x}_t = (x_t^1, \ldots, x_t^N)$ and scene-specific physical properties.

3   Estimate initial state $\hat{\mathbf{x}}_{t=1} = (\hat{x}_1^1, \ldots, \hat{x}_1^N)$ as described in Sec. 4.2.

4   Warmup $\mathcal{R}_\phi$ using rendering loss given $\hat{\mathbf{x}}_{t=1}$ on multi-view observation on $t = 1$.

5   Freeze $\mathcal{T}_\theta, \mathcal{R}_\phi$.

6   // Stage B: Optimize scene-specific latent features as visual posteriors

7   Initialize random Gaussian distribution $\mathcal{N}(\hat{\mu}^i, \hat{\sigma}^i)$ for each particle $i$.

8   **while** *not converged* **do**

9      **for** *time step* $t = 1 \ldots T$ **do**

10         Sample particle feature for each particle: $\hat{z}_t^i \sim \mathcal{N}(\hat{\mu}^i, \hat{\sigma}^i)$ and produce $\hat{\mathbf{z}} = (\hat{z}^1, \ldots, \hat{z}^N)$.

11         Simulate the next step $\hat{\mathbf{x}}_t = \mathcal{T}_\theta(\hat{\mathbf{x}}_{t-1}, \hat{\mathbf{z}})$ .

12         Sample camera rays $\mathbf{r} \in R(\mathbf{P})$ and predict $\hat{\mathbf{C}}(\mathbf{r}, t)$ as described in Sec. 4.2.

13         Update the posterior distribution $\mathcal{N}(\hat{\mu}^i, \hat{\sigma}^i)$ for each particle $i$ using rendering loss $\mathcal{L}_t$ (Sec. 4.2).

14   // Stage C: Finetune the pretrained physical prior learner $p_\psi$

15   Fix distribution $\mathcal{N}(\hat{\mu}^i, \hat{\sigma}^i)$ for each particle $i$.

16   **while** *not converged* **do**

17      **for** *time step* $t = 1 \ldots T$ **do**

18         Sample particle feature for each particle: $\tilde{\mathbf{z}}_t \sim p_\psi(\tilde{\mathbf{z}}_t \mid \hat{\mathbf{x}}_{1:t-1}, \tilde{\mathbf{z}}_{t-1})$.

19         Simulate the next step $\hat{\mathbf{x}}_t = \mathcal{T}_\theta(\hat{\mathbf{x}}_{t-1}, \tilde{\mathbf{z}}_t)$ .

20         Predict $\hat{\mathbf{C}}(\mathbf{r}, t)$ by volume rendering as described in Sec. 4.2.

21         Update the prior learner $p_\psi$ using the loss function $\mathcal{L}_\psi$ in Eq. 3.

22   Simulate novel scenes with the learned $\mathcal{T}_\theta$ and $p_\psi$.

---

## C DATASETS

### C.1 PARTICLE DATASETS

This dataset is used for pretraining the probabilistic fluid simulator in Stage A. Following (Ummenhofer et al., 2020; Prantl et al., 2022), we simulate the dataset with DFSPH (Bender & Koschier, 2015) using the SPlisHSPlasH framework[2]. This simulator generates fluid flows with low-volume compression. The particle dataset contains 600 scenes, with 540 scenes used for training and 60 scenes reserved for the test set. In each scene, the simulator randomly places a fluid body of random shape in a cubic box (see *Default Boundary* in Figure 9) and the fluid body freely falls under the influence of gravity and undergoes collisions with the boundary. The initial fluid body in each scene is applied with random rotation, scaling, and initial velocity. The simulator randomly samples physical properties viscosity $\nu$ and density $\rho$ for fluid bodies from uniform distribution $\mathcal{U}(0.01, 0.4)$ and $\mathcal{U}(500, 2000)$ respectively. The simulation lasts for 4 seconds, which consists of 200 time steps. In general, there are $273 \sim 19682$ fluid particles in each scene in this dataset.

### C.2 GENERATION OF VISUAL OBSERVATIONS

The visual observations are used in Stages B & C. Under each set of physical parameters, we generate a single 3D video of fluid dynamics with *Cuboid* (unseen during Stage A) geometry. Each example contains 60 time steps, where the most significant dynamic changes are included. We use Blender (Community, 2018) to generate multi-view visual observations. Each fluid dynamic example is captured from 20 randomly sampled viewpoints, with the cameras evenly spaced on the upper hemisphere containing the object. The first 50 time steps are used for training, and the last 10 time steps are used for the evaluation of future prediction.

### C.3 EVALUATION SETS OF VISUAL PHYSICAL INFERENCE

Our model and the baselines are evaluated on 3 challenging novel scene simulation tasks. Figure 9 shows boundaries and fluid geometries used for evaluation.

- **Unseen Fluid Geometries.** We use *Standford Bunny*, *Sphere*, *Dam Break* that are unseen during pretraining as fluid bodies. The default boundary used in Stage A is used as a fluid container. We generate 50 sequences of 60 time steps under each set of physical parameters for evaluation. In each evaluation scene, random fluid geometry is chosen and random rotation, scaling, and velocities are imposed on the fluid body.

- **Unseen Boundary conditions.** We use *Dam Break* fluid and unseen boundary for this task. This evaluation set features an out-of-domain boundary with a slender pillar positioned in the center. In each scene, the *Dam Break* fluid collapses and strikes the pillar in the container, which was unseen during pretraining and from visual observations. Similarly, random rotation, scaling, and velocities are imposed on *Dam Break* fluid, and 50 sequences of 60 time steps under each set of physical parameters for evaluation.

The horizontal initial velocity of the fluid body is randomly sampled from a uniform distribution of $\mathcal{U}(-2, 2)$, with a vertical initial velocity of 0, and scale by a factor randomly sampled from another uniform distribution of $\mathcal{U}(0.8, 1.2)$. Table 9 shows the particle number of fluid bodies with no scaling applied. Table 8 illustrates the features of the visual observation scene and evaluation sets.

### C.4 DYNAMICS DISCREPANCIES

In this task, we simulate two heterogeneous fluids with different physical properties. The generation of visual examples of this task is the same as previous tasks, but with two fluids of different physical properties interacting with each other. The evaluation set contains the same heterogeneous fluids but with unseen fluid geometries as fluid bodies. The evaluation set contains 50 sequences in total. Similarly, random rotation, scaling, and velocities are imposed on the fluid bodies in scenes in the evaluation set.

---

[2]https://github.com/InteractiveComputerGraphics/SPlisHSPlasH

Table 8: The features of the visual observation scene and evaluation sets. *Visual* means the visual observation scene, *Geometry* means evaluation sets of unseen fluid geometries, and *Boundary* means evaluation sets of unseen boundary conditions.

| | Boundary | Fluid Body | Rotation, Scaling, Velocities | #Sequence | Time Steps |
|---|---|---|---|---|---|
| Visual | Default | Cuboid | - | 1 | 60 (50 for training ) |
| Geometry | Default | Standford Bunny, Sphere, Dam Break | ✓ | 50 | 60 |
| Boundary | Unseen | Dam Break | ✓ | 50 | 60 |

Table 9: Particle number of different fluid bodies with no scaling applied.

| Dataset | Cuboid | Stanford Bunny | Sphere | Dam Break |
|---|---|---|---|---|
| Particle Numbers | 6144 | 8028 | 8211 | 10368 |

# D  EXPERIMENTAL DETAILS

## D.1  EVALUATION METRICS

We adopt the evaluation metric from the literature of CConv Ummenhofer et al. (2020). For all experiments but the short-term predictions in Table 3, we compute the average Euclidean distance from the true particle positions $(p_t^i)$ to the positions of the closest predicted particles $(\hat{p}_t)$:

$$d = \frac{1}{T \times N} \sum_t \sum_i \min_{\hat{p}_t} ||p_t^i - \hat{p}_t||_2$$

where $T$ is the prediction time horizon and $N$ is the number of particles. In particular, for the short-term prediction experiments ($n+1$ and $n+2$) in Table 3, we compute an one-to-one mapping metric, the average error of the predicted particles w.r.t. the corresponding ground truth particles:

$$d_t = \frac{1}{N} \sum_i ||p_t^i - \hat{p}_t^i||_2$$

where $\hat{p}_t^i$ is the predicted position for particle $i$.

## D.2  VISUAL PHYSICAL INFERENCE

### D.2.1  BASELINES

Our method is compared with the four baseline models below. For NeuroFluid and system identification, we adopt PhysNeRF with the same architecture and hyperparameter in Table 7. All models are trained given the single 3D video of fluid dynamics with the *Cuboid* geometry and evaluated on novel scenes given ground truth initial states.

- **CConv (Ummenhofer et al., 2020):** Given the initial particle positions and velocities, CConv (Ummenhofer et al., 2020) uses a continuous convolution network that is performed in 3D space to simulate the particle transitions. However, the input to the CConv model only contains particle position and velocity such that it has a limitation that it can only perform simulation on fluid with identical physical parameters. We use this CConv model which has no additional input feature, pretrained on particle dataset in Sec. C.1 and directly evaluate this model on novel scene simulations.

- **NeuroFluid (Guan et al., 2022):** A fully differentiable method for fluid dynamics grounding that links particle-based fluid simulation with particle-driven neural rendering in an end-to-end trainable framework. This approach links particle-based fluid simulation with particle-driven neural rendering in an end-to-end trainable framework, such that the two networks can be jointly optimized to obtain reasonable particle representations between them. We adopt the CConv model pretrained on the particle dataset in Sec. C.1 as the initial transition model and optimize NeuroFluid on *Cuboid* scene in an end-to-end manner.

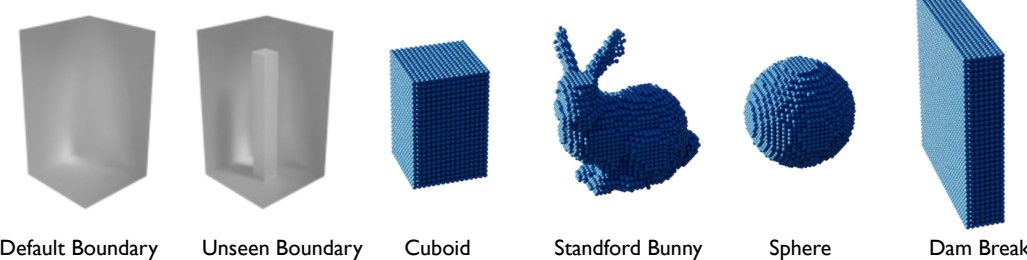

Default Boundary    Unseen Boundary    Cuboid    Standford Bunny    Sphere    Dam Break

Figure 9: Visualization of used boundaries and fluid bodies in visual physical inference. The leftmost container is the default boundary used in each scene in particle datasets and *Cuboid*.

- **PAC-NeRF (Li et al., 2023):** PAC-NeRF is a method that estimates physical parameters from multi-view videos with Eulerian-Lagrangian representation of neural radiance field and MPM simulator. We feed the physical parameters estimated on *Cuboid* scene to the MPM simulator and use the given physical parameters to rollout on novel scenes.

- **System Identification (Sys-ID):** For system identification, we train another CConv but with additional features of physical parameters for input including viscosity $\nu$ and density $\rho$. Sys-ID optimizes fluid properties by backpropagating the rendering loss through the trained transition network and PhysNeRF. The optimized physical parameters are used as model inputs to simulate novel scenes.

Note that baselines and our model are optimized over the entire visual sequence.

### D.2.2 IMPLEMENTATION DETAILS

We train our model and the baselines with multi-view observations on the fluid sequence of the *Cuboid* geometry. Before Stage B, The PhysNeRF is finetuned on visual observations for $100k$ steps with learning rate $3\mathrm{e}{-}4$ and exponential learning rate decay $\gamma = 0.1$ given the estimated initial state and multi-view observation of the first frame. After that, we freeze $\mathcal{R}_\phi$ and $\mathcal{T}_\theta$ (pretrained on Particle Dataset) and infer the visual posterior by backpropagating the rendering loss. Then the physical prior learner $p_\psi$ is trained to adapt to the inferred visual posterior. The visual posterior latent and physical prior learner are separately optimized for $100k$ steps and $50k$ steps in Stage B and Stage C, with a learning rate of $1\mathrm{e}{-}4$ and a cosine annealing scheduler.

### D.2.3 INITIAL STATE ESTIMATION

We estimate the initial state of fluid particles given multi-view observation on the first frame and then feed the estimated initial state to all simulators and the neural renderer. We estimate the initial particle positions using the voxel-based neural rendering technique (Liu et al., 2020; Sun et al., 2022; Müller et al., 2022) and maintain an occupancy cache to represent empty *vs.* nonempty space. At test time, we randomly sample fluid particles within each voxel grid to generate initial particle positions. To maintain spatial consistency between the estimated initial particle positions and the particle density generated by SPlisHSPlasH (Bender et al., 2022), we employ the fluid particle discretization tools provided in SPlisHSPlasH to calculate the fluid particle count per unit volume. When estimating initial particle positions with new fluid scenes, we adopt the voxel-based neural renderer to predict the spatial volume occupied by the fluid, enabling the subsequent sampling of fluid particles according to the prescribed fluid particle density per unit volume. Since we do not apply random initial velocities to the fluid dynamics in visual examples, the initial velocities are set as zero.

### D.3 EVALUATION OF PROBABILISTIC FLUID SIMULATOR

#### D.3.1 BASELINES ON PARTICLE DATASET

We compare our probabilistic fluid simulator with four representative particle simulation approaches, based on GNN, continuous convolution, and Transformer, *i.e.*, DPI-Net (Li et al., 2019), CConv (Ummenhofer et al., 2020), DMCF (Prantl et al., 2022), and TIE (Shao et al., 2022).

- **DPI-Net (Li et al., 2019):** DPI-Net is a particle-based simulation method that combines multi-step spatial propagation, a hierarchical particle structure, and dynamic interaction graphs.

- **CConv (Ummenhofer et al., 2020):** The method employs spatial convolutions as the primary differentiable operation to establish connections between particles and their neighbors. It predicts particle features and dynamics in a smooth and continuous way.

- **DMCF (Prantl et al., 2022):** DMCF imposes a hard constraint on momentum conservation by employing antisymmetrical continuous convolutional layers. It utilizes a hierarchical network architecture, a resampling mechanism that ensures temporal coherence.

- **TIE (Shao et al., 2022):** A Transformer-based model that captures the complex semantics of particle interactions in an edge-free manner. The model adopts a decentralized approach to computation, wherein the processing of pairwise particle interactions is replaced with per-particle updates. The original method is conducted on the PyFlex dataset with less number of fluid particles ($\sim$ hundreds). However, on *Particle Dataset* ($\sim$ thousands), the experiment on TIE leads to unacceptable memory cost. Therefore, we downsample the fluid particles in each scene of the *Particle Dataset* to the ratio of $1/20$.

#### D.3.2 IMPLEMENTATION DETAILS

In Stage A, the probabilistic fluid simulator is trained to predict two future states from two inputs. The ADAM optimizer (Kingma & Ba, 2015) is used with an initial learning rate of $0.001$ and a batch size of 16 for $50k$ iterations. We follow previous works (Ummenhofer et al., 2020; Prantl et al., 2022) to set a scheduled learning rate decay where the learning rate is halved every $5k$ iterations, beginning at iteration $25k$. The latent distribution of each particle is an $8$-dimensional Gaussian with parameterized mean and standard deviation. The KL regularizer $\beta$ is set as $0.1$, shown in Table 7. The experiments are conducted on 4 NVIDIA RTX 3090 GPUs. To enhance long-term prediction capability, the probabilistic fluid simulator is trained to predict 5 future states from 5 inputs for experiments in visual physical inference. Additionally, we use $\tanh$ as an activation function for layers in the shared feature encoder (CConv). This is to ensure the learned posterior of latent distribution from visual observation lies in the space of the learned physical prior learner.

## E EXPERIMENTS WITH NON-ZERO INITIAL VELOCITIES OF THE OBSERVED VISUAL SCENE

To assess the performance of our method in a more general case, specifically, learning from visual observations of fluids with non-zero initial velocities, we modify the training scheme in Stage B by randomly initializing $\{v_{t=1}^i\}_{i=1:N}$ and treating them as trainable parameters. The optimization of $\{v_{t=1}^i\}_{i=1:N}$ is carried out concurrently with the optimization of the visual posteriors $\hat{\mathbf{z}}$. We evaluate the results against a model trained with true non-zero initial velocities given on the observed scene. In Table 10, we compare future prediction errors on observed scenes of fluids with non-zero initial velocities. In Table 11, we compare the average simulation errors of the two models on novel scenes with new initial geometries and boundaries. We can observe that both models produce comparable results, showcasing the ability of our method to infer uncertain initial velocities of fluids by treating them as optimized variables.

Table 10: Quantitative results of future prediction error $d$ on observed scenes of fluids with non-zero initial velocities. We compare the performance of the model trained with optimized velocity against the model with ground truth non-zero initial velocity given.

| METHOD | $\rho = 2000, \nu = 0.065$ | $\rho = 1000, \nu = 0.08$ | $\rho = 500, \nu = 0.2$ |
|---|---|---|---|
| TRAINED WITH TRUE VELOCITY | **39.02** | **39.14** | **43.57** |
| TRAINED WITH OPTIMIZED VELOCITY | 39.53 | 40.99 | 45.31 |

Table 11: Average prediction error $d$ on novel scenes with new initial geometries and boundaries (denoted by Geom. and Bdy.). Each model is trained with a single visual observation with non-zero velocity on three physical property sets.

| | $\rho = 2000, \nu = 0.065$ | | $\rho = 1000, \nu = 0.08$ | | $\rho = 500, \nu = 0.2$ | |
|---|---|---|---|---|---|---|
| METHOD | GEOM. | BDY. | GEOM. | BDY. | GEOM. | BDY. |
| TRAINED WITH TRUE VELOCITY | 33.39 | 39.74 | **33.50** | **38.41** | 38.52 | 47.23 |
| TRAINED WITH OPTIMIZED VELOCITY | **33.25** | **39.55** | 33.72 | 38.75 | **38.34** | **46.90** |

## F  ADDITIONAL VISUALIZATION RESULTS

Figure 10, 11, & 12 provide visualizations of the predicted particles of baselines and our model on the novel scenes with unseen fluid geometries. Figure 13, 14, & 15 provide visualizations of the predicted particles of baselines and our model on the novel scenes with unseen boundary conditions. We can see that CConv and NeuroFluid tend to produce noisy predictions on novel scenes. PAC-NeRF produces sporadic predictions that are contrary to the continuous dynamics behavior as ground truth simulations. Our model has more reasonable prediction results and can effectively respond to different physical parameters.

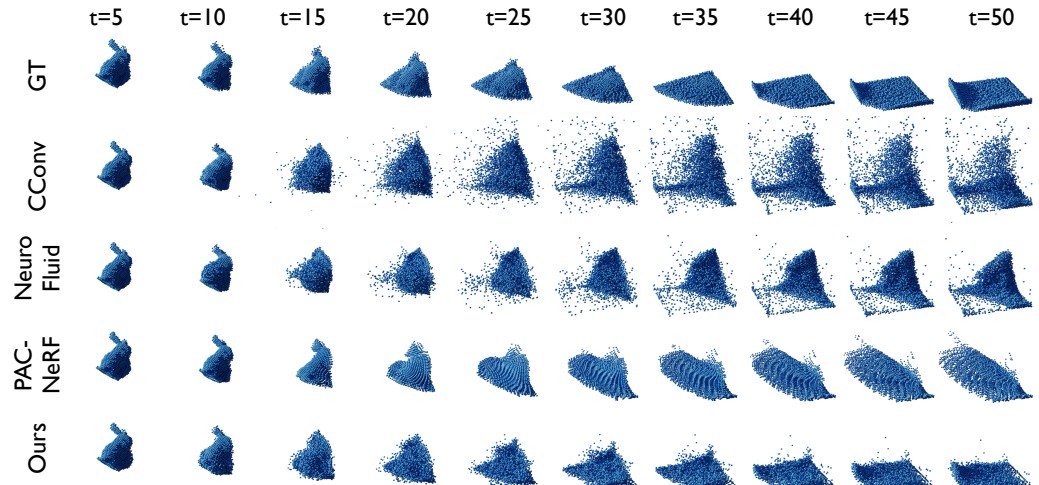

Figure 10: Qualitative results of simulated particles from learned simulators on the unseen fluid geometry (Stanford Bunny) with physical parameter $\rho = 2000, \nu = 0.065$. In this scenario, the *Stanford Bunny* fluid is randomly sampled with a relatively high horizontal initial velocity and initially makes contact with the side of the container. Due to its low viscosity, the fluid flows rapidly down along the container's side.

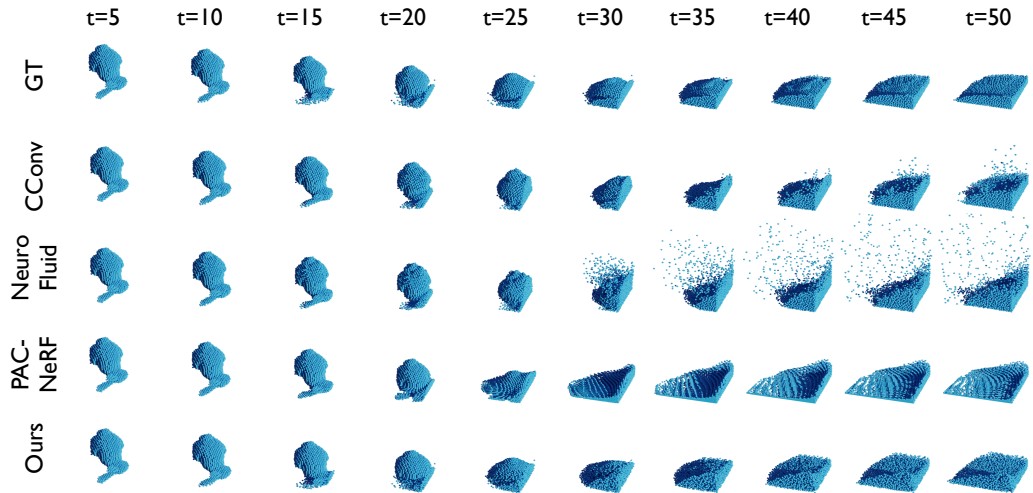

Figure 11: Qualitative results of simulated particles from learned simulators on the unseen fluid geometry (Stanford Bunny) with physical parameter $\rho = 1000, \nu = 0.08$. In this scenario, the *Stanford Bunny* fluid apply to random rotation, presenting an inverted initial state, and is launched into the container with a relatively low horizontal initial velocity in a projectile motion.

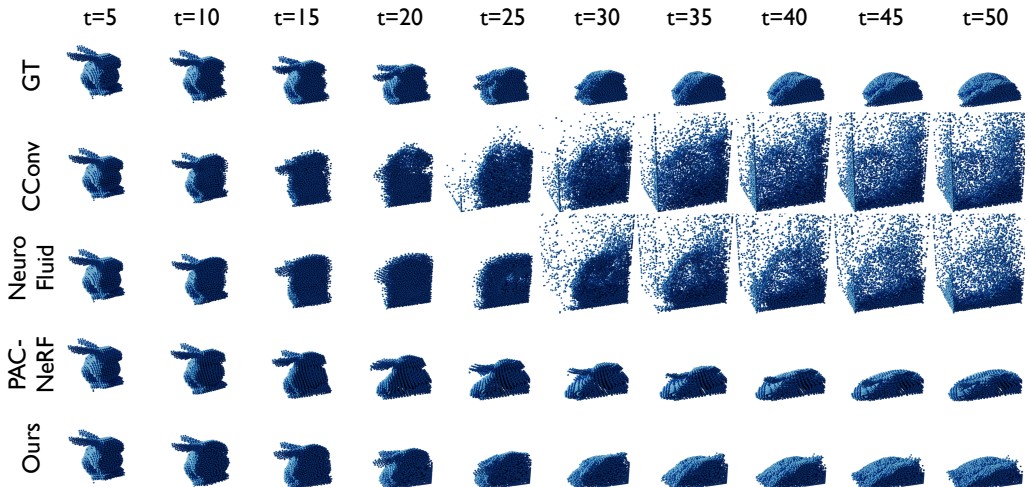

Figure 12: Qualitative results of simulated particles from learned simulators on the unseen fluid geometry (Stanford Bunny) with physical parameter $\rho = 500, \nu = 0.2$. In this scenario, the *Stanford Bunny* fluid is randomly sampled with a relatively high horizontal initial velocity and makes initial contact with the side of the container. Due to its high viscosity, the fluid slowly flows down along the container's side.

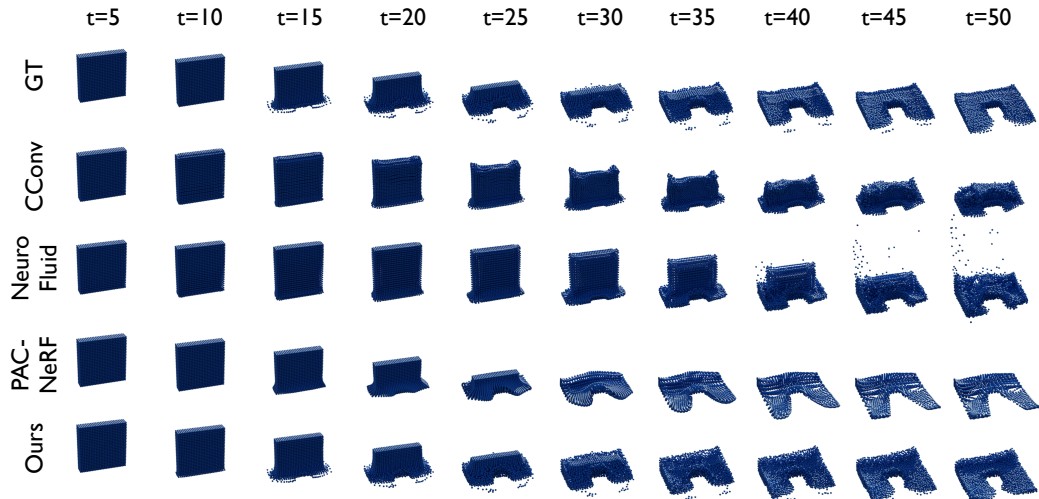

Figure 13: Qualitative results of simulated particles from learned simulators on the unseen boundary with physical parameter $\rho = 2000, \nu = 0.065$. In *unseen boundary* scenarios, We use the boundary with a slender pillar positioned in the center. In each scenario, the Dam Break fluid collapses and strikes the pillar in the container, which was unseen during pretraining and from visual observations.

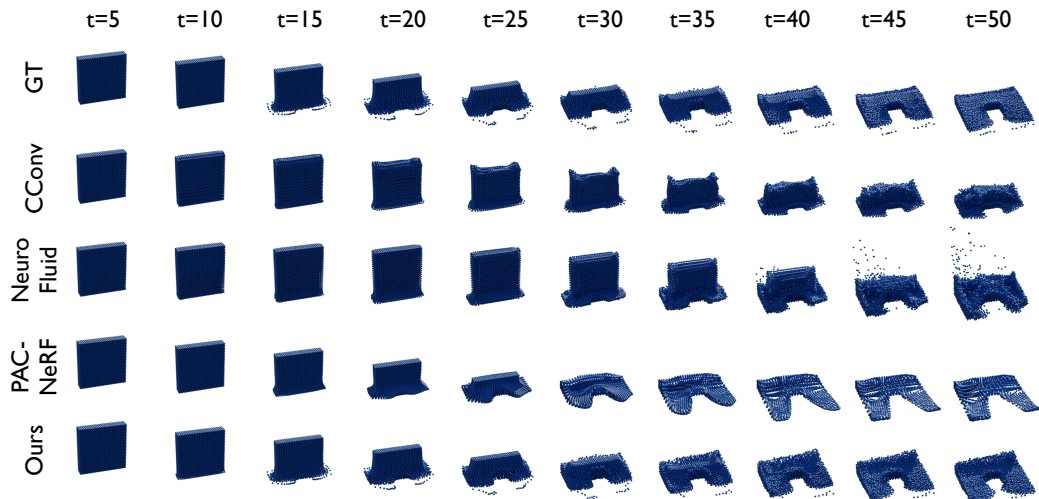

Figure 14: Qualitative results of simulated particles from learned simulators on the unseen boundary with physical parameter $\rho = 1000, \nu = 0.08$. PAC-NeRF produces discontinuous simulation predictions.

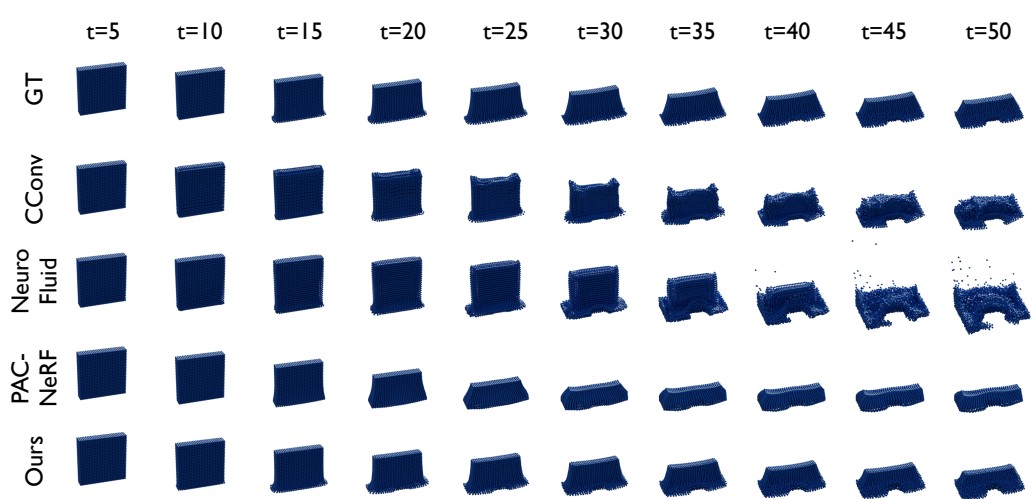

Figure 15: Qualitative results of simulated particles from learned simulators on the unseen boundary with physical parameter $\rho = 500, \nu = 0.2$.

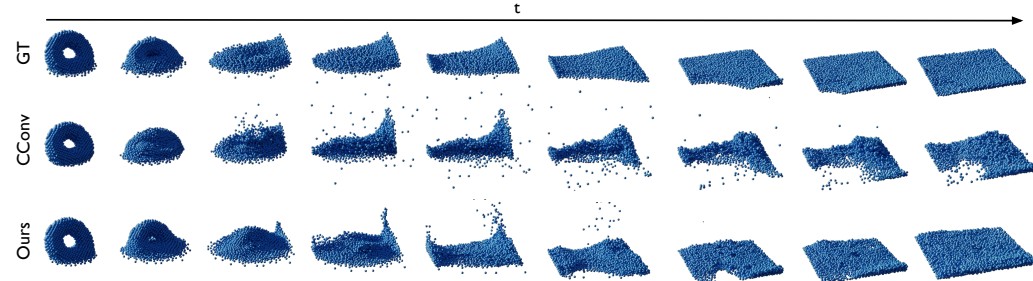

Figure 16: Qualitative results of predicted long-term simulation on the pretrained particle dataset.

We present qualitative results of the predicted particles of the pretrained probabilistic fluid simulator in Figure 16. We can see that our method produces closer prediction results with ground truth. This indicates that our probabilistic fluid simulator shows the capability to capture broad hidden physics within the latent space, leveraging spatiotemporal correlation in particle data.

## G  FURTHER DISCUSSIONS ON REAL-WORLD EXPERIMENTS

The primary focus of this paper is to explore the feasibility of the proposed inference–transfer learning scheme for physics, we first use synthetic visual data to evaluate the proposed method. In real-world experiments, however, we need to use specialized fluid flow measurement techniques, such as Particle Image Velocimetry (PIV), to measure the performance of the model. Most inverse graphics methods are not applied to real-world scenarios of fluid dynamics that undergo intense change and visual noises, including prior arts like NeuroFluid (Guan et al., 2022) and PAC-NeRF (Li et al., 2023). The visual observations of fluid dynamics in real-world scenarios contain more visual noises, such as reflection and refraction, which makes it harder to cope with.

Nevertheless, we acknowledge that real-world validation is meaningful and challenging. To explore the possibility of applications of *latent intuitive physics* in real-world scenarios with complex and noisy visual observation, we made our best efforts to conduct real-world experiments. As the pipeline provided in Figure 17, we capture the dynamics of dyed water in the fluid tanks. We collect RGB images at a resolution of $1,200 \times 900$ on the hemisphere of the scene. As the presence of refraction and reflection phenomena (which are usually absent in other dynamics contexts) will bring extra burden for the reconstruction of fluid geometries and dynamics, we first adopt NeRFREN (Guo et al., 2022) to remove the inherent reflection and refraction of fluids and the container. Then we segment the fluid body and remove the background using SAM (Kirillov et al., 2023). The preprocessed images are then used to estimate fluid positions using the initial state estimation module. Estimated positions are shown in Figure. 17. Please refer to the supplementary video for a vivid illustration of real-world experiments. However, an outstanding challenge of the complete experiment is to acquire high frame-rate images with synchronized cameras across multiple viewpoints. Therefore, we have to leave this part for future work.

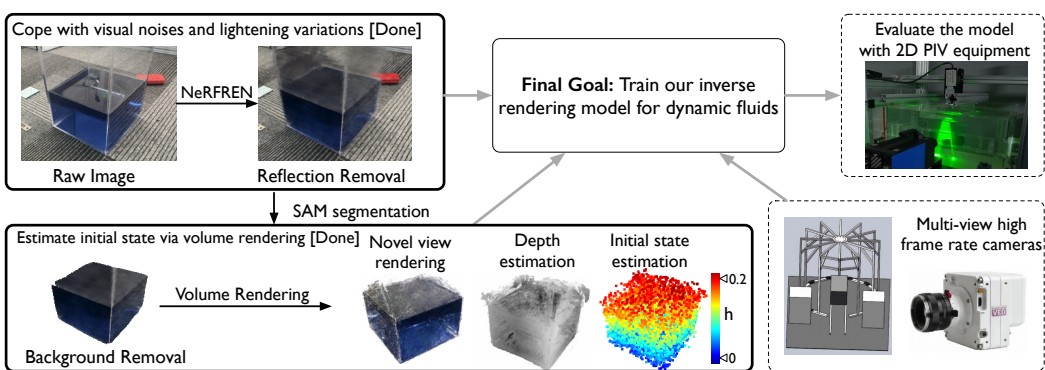

Figure 17: Our pipeline and intermediate results for real-world experiments. Due to the requirements of advanced data acquisition equipment, we leave the complete experiment of dynamic scenes to future work (dashed boxes).

