# OpenReview forum: "Latent Intuitive Physics: Learning to Transfer Hidden Physics from A 3D Video"
_ICLR.cc/2024/Conference — ICLR 2024 poster_

### Official Review · Reviewer_eXxy · 2023-10-29

**Soundness:** 3 good
**Presentation:** 2 fair
**Contribution:** 3 good
**Rating:** 6
**Confidence:** 4

**Summary:**

This paper proposes to transfer the attributes of fluid from videos for further simulations in novel scenes. The properties shown in videos are encoded into a latent space, which is further used by pretrained probabilistic physics engine to rolloout the dynamics of fluid in unseen scenes. Experiments show the effectiveness of probabilistic simulation models over baselines as well as the superior performance of physical inference from visual observations.

**Strengths:**

* A probabilistic simulation model with better performance than existing models.
* The proposed method is able to transfer the properties from videos to simulate the fluid dynamics in novel scene.
* The proposed method obtains superior performance in most cases.

**Weaknesses:**

1. Since this is a probabilistic model, the predicted dynamics may be different for different sampled feature $\mathbf{z}_t$. Is there any reference to show the variations of predicted particle states? For example, the model would predict one sequence for multiple times and calculate the standard deviation compared with the ground truth.
2. What's the unit of measurement in Table 3? Is it millimeter or centimeter?
3. What's the standard deviation of the prediction errors in Table 2? For example, one scene should include several sequences. The errors may vary from sequence to seqence. What's the variation of the errors?

**Questions:**

* In Fig 4, is it an accurate reference of the results for the phrase "row 3-6". It seems like the results start from row 2.

---

> ### Author Response · Authors · 2023-11-19
> **Responses to Reviewer eXxy**
>
> Thank you for your comments. We have made every effort to answer each of your questions and hope that our response can address your concerns.
>
> > Q1: Is there any reference to show the variations of predicted particle states?
>
> Good question! It is crucial to understand the diversity in probabilistic dynamics prediction. Therefore, in Tables 1~4 in the revised paper, we predict each sequence 10 times with different $\mathbf z_t$ drawn from the prior module, and calculate the standard deviation of the errors. The results are shown below:
>
> Our results for novel scene simulation ($t\in [2,60]$), added to Table 1:
> | Setting      | $\rho=2000,\nu=0.065$ | $\rho=1000,\nu=0.08$ | $\rho=500,\nu=0.2$  |
> | ------------ | --------------------- | -------------------- | ------------------- |
> | New geometry | 34.54 $\pm$ 0.55 | 33.11 $\pm$ 0.50  | 39.03 $\pm$ 0.79 |
> | New boundary | 39.86 $\pm$ 0.90 | 37.79 $\pm$ 0.84  | 47.25 $\pm$ 1.26 |
>
> Our future prediction results for the observed scenes ($t\in [51,60]$):
> | $\rho=2000,\nu=0.065$ | $\rho=1000,\nu=0.08$                         | $\rho=500,\nu=0.2$ |
> | ---------------------- | -------------------------------------------- | ------------------ |
> | 32.41 $\pm$ 0.17    | 32.97 $\pm$ 0.71 |         41.15 $\pm$ 0.71           |
>
> Our results on particle datasets, added to Table 3:
> | $d_{t+1}$                | $d_{t+2}$            | Average error over 190 steps |
> | ------------------ | ----------------- | ---------------------------- |
> | 0.31 $\pm$ 0.003 | 0.94 $\pm$ 0.011 | 38.37 $\pm$ 0.860             |
>
> Our results of prediction error on generalization to unseen fluid dynamics, added to Table 4:
> |                 | Observed               | Unseen |
> | ------------------ | ----------------- | ---------------------------- |
> | Ours | 36.03 $\pm$ 0.80 | 44.25 $\pm$ 1.36            |
>
> As we can see from the updated results, the predictions of our model yield larger variances as the prediction time horizon extends.
>
> Note: In this context, we conduct multiple simulations to compute the standard deviation for each individual sequence. This differs from our evaluation setup for the subsequent question, where we "calculate the standard deviation of prediction errors across multiple sequences".
>
> > Q2: What's the unit of measurement in Table 3? Is it millimeter or centimeter?
>
> All the tables in our paper, including Table 3, use millimeters as the unit of measurement.
>
> > Q3: What's the standard deviation of the prediction errors in Table 2? For example, one scene should include several sequences. The errors may vary from sequence to sequence.
>
> First, we want to clarify that Table 2 showcases the future prediction errors in observed scenes, where only videos of **a single sequence** are provided for each physical parameter.
>
> Instead, in Table 1, we do evaluate our method on 50 different sequences with diverse unseen fluid geometries and boundary conditions for each physical parameter. Below, we present the standard deviation of the simulation errors across these sequences.
>
> We should note that there are substantial variations in simulation difficulty between these samples, resulting from diverse initial conditions (including initial positions and velocities) and boundary conditions. Consequently, we observe large standard deviations for all the compared methods. Despite these challenges, our model consistently presents significantly lower average errors and the lowest inter-sample standard deviation as well. This demonstrates its robustness in effectively handling diverse and challenging fluid scenes.
>
> |            | $\rho=2000,\nu=0.065$ (Geom. / Bdy.) | $\rho=1000,\nu=0.08$ (Geom. / Bdy.)   | $\rho=500,\nu=0.2$ (Geom. / Bdy.)     |
> | ---------- | ------------------------------------- | ------------------------------------- | ------------------------------------- |
> | CConv      | 52.49 $\pm$ 29.83 / 64.29 $\pm$ 56.95 | 51.33 $\pm$ 33.47 / 56.33 $\pm$ 44.12 | 40.67 $\pm$ 21.12 / 53.28 $\pm$ 30.74 |
> | NeuroFluid | 65.01 $\pm$ 45.16 / 73.55 $\pm$ 64.95 | 59.79 $\pm$ 46.17 / 60.46 $\pm$ 48.35 | 40.88 $\pm$ 21.20 / 50.73 $\pm$ 27.94 |
> | PAC-NeRF   | 51.10 $\pm$ 31.39 / 59.61 $\pm$ 43.80 | 51.33 $\pm$ 26.78 / 56.84 $\pm$ 37.11 | 40.97 $\pm$ 22.07 / 62.05 $\pm$ 45.85 |
> | Ours       | 32.88 $\pm$  7.30 / 39.86 $\pm$ 12.32 | 32.60 $\pm$ 7.58 / 37.62 $\pm$ 14.60  | 39.29 $\pm$ 14.70 / 47.31 $\pm$ 23.58 |
>
> > Q4: In Fig 4, is it an accurate reference of the results for the phrase "row 3-6"? It seems like the results start from row 2.
>
> Thank you for your careful review. It's a typo and we have corrected it to "Rows 2-6" in the revision.

---

### Official Review · Reviewer_zRiM · 2023-10-29

**Soundness:** 3 good
**Presentation:** 3 good
**Contribution:** 3 good
**Rating:** 6
**Confidence:** 1

**Summary:**

This paper proposes a new learning framework to extract hidden physical properties from a single 3D video to simulate the observed fluid in new scenes. It works in a probabilistic setting with a particle-based model. A parameterized prior gets the visual observations as input and tries to approximate a 'visual posterior' obtained from a learned neural renderer. Numerical results demonstrate the effectiveness of the proposed approach in novel scene simulations and the prediction of the future behavior of the observed fluid.

**Strengths:**

I have little background in the main areas of this publication. The overall method appears to be sophisticated, well-designed and sufficiently different from related work as far as I can judge.  The numerical experiments make a solid impression. They test different aspects of the proposed method against competitors, give promising results, ablate at least some design choices (of including stages B and C in the framework), and briefly discuss the extension from simulated to real-world experiments.

**Weaknesses:**

I found the paper very dense and extremely hard to follow for someone outside of the field (like myself). A more introductory section including a more formal (mathematical) definition of which functions with what kind of inputs and outputs are approximated by certain networks would have helped me.  Similarly, the description of the experiments and their evaluation could have been clearer including the metrics used for the evaluation (Tables 2 and 3 just say "prediction errors", Table 4 does not specify a metric at all, and Table 1 refers to "average position errors on unseen fluid geometries and boundary conditions", which I cannot understand either). Yet, such aspects might be clear to anyone more familiar with the topic. As I am neither familiar with graphics, nor with particle-based simulations, nor with learned probabilistic models, I am certainly not a good reviewer for this paper (and I informed the area chair accordingly).

**Questions:**

I fear I'd have to invest significant extra time in reading this (but also prior) work in order to ask meaningful questions.

---

> ### Author Response · Authors · 2023-11-19
> **Responses to Reviewer zRiM**
>
> We appreciate your efforts to review our work. In addition to the following responses, we have also made comprehensive revisions to the paper to enhance its overall clarity.
>
> > Q1: I found the paper very dense and extremely hard to follow for someone outside of the field.
>
> We apologize for any difficulty you experienced in reading our paper. We have tried our best to organize the writing, but found it challenging due to the complexity of the proposed method, which involves multiple training stages and many details. To improve the overall reading experience, we have made the following changes:
> - We refined Figure 3 to include all network components and clarify the inputs and outputs of each module.
> - We introduced Figure 7 in Appendix A.1, giving the details of each training stage.
> - Moreover, Table 6 now summarizes each component's formulations, inputs, outputs, active training stages, and objective functions.
>
> We hope these modifications will improve the readability of the paper and provide a clearer understanding of our approach. If you have any further suggestions or concerns, please feel free to let us know.
>
> > Q2: The description of the experiments and their evaluation could have been clearer including the metrics used for the evaluation.
>
> For all experiments but the short-term predictions in Table 3, we compute the average Euclidean distance from the true particle positions ($p_t^i$) to the positions of the closest predicted particles ($\hat{p}_t$):
>
> $$\bar{d}=\frac{1}{T \times N} \sum_t \sum_i \min _{\hat{p}_t} ||p_t^i - \hat{p}_t||_2$$ where $T$ is the prediction time horizon and $N$ is the number of particles. As mentioned in Section 5.1 ("Settings"), we adopt this evaluation metric from the previous work by Ummenhofer et al. (2020).
>
> In particular, for the particle-based experiments for short-term prediction in Table 3 (the first two columns), we compute a one-to-one mapping metric, the average error of the predicted particle positions w.r.t. the true positions of corresponding particles:
>
> $$d_t=\frac{1}{N} \sum_{i}  ||p_t^i - \hat{p}_t^i||_2$$ where $\hat{p}_t^i$ is the predicted position for particle $i$.

---

### Official Review · Reviewer_QGTH · 2023-10-31

**Soundness:** 3 good
**Presentation:** 2 fair
**Contribution:** 3 good
**Rating:** 6
**Confidence:** 3

**Summary:**

In this manuscript, the authors investigated a transfer learning framework for physics simulation from 3D Video. While conventional methods require known physical states to infer fluid simulation, this study has developed a framework that operates without them, relying on latent features extracted from 3D video inputs.

**Strengths:**

I find this paper interesting and agree that getting inspiration from the concept of intuitive physics plays an essential role in the literature. In addition, modeling a model in terms of video-based processing can contribute to future research on understanding human brain mechanisms.

**Weaknesses:**

This framework includes many model processing, and it was hard for me to understand the whole structure at the first read. In particular, when I tried to understand the model architecture, I expected Figure 3 to include all the information. However,  Figure 3 lacks information about the neural renderer, and the term "type-aware preprocessing" is not used in the main text, leading to comprehension difficulties. Improving the connection between Figures 2 and 3 would enhance readers' understanding.

**Questions:**

I expect the authors' future direction of real-world videos. It might be beneficial to describe the current state and the challenges more in the main text in addition to the information provided in the Appendix.

---

> ### Author Response · Authors · 2023-11-19
> **Responses to Reviewer QGTH**
>
> Thank you for your valuable comments. We hope that the responses below can fully address your concerns. If you have any additional suggestions, please don't hesitate to inform us.
>
> > Q1: This framework includes many model processing, and it was hard for me to understand the whole structure at the first read.
>
> (1) We apologize for any difficulty you experienced in reading our paper. We have made the following changes to improve readability:
> - Following your suggestion, we refined **Figure 3** to involve all network components of the model architecture, including the neural renderer. Specifically, the neural renderer follows the implementation from Guan et al. (2022), which generates separate view-independent and view-dependent encodings based on predicted particle positions. A detailed description of PhysNeRF is included in Appendix A.3.
> - In Appendix A.1, we added a new figure (**Figure 7**) to illustrate the training details of the particle-space pretraining stage, the visual inference stage, and the prior adaptation stage, respectively. Moreover, we have improved **Table 6** to give a more detailed summarization of each component's formulations, inputs, outputs, active training stages, and objective functions.
>
> (2) Regarding "type-aware preprocessing" in the original Figure 3:
>
> The "type-aware preprocessing" follows the implementation and the terminology from the CConv paper (Ummenhofer et al., 2020). As depicted in Figure 2 of their paper, this mechanism is realized with two separate CConv layers: One is to encode the *fluid particles* in the neighborhood of each particle location; The other one is to handle the *virtual boundary particles* in the same neighborhood. The features extracted from these processes are then concatenated to form the inputs for subsequent layers. We have provided a detailed description of how the model processes fluid and boundary particles in Appendix A.2. To enhance the clarity of Figure 3, we have omitted the "type-aware preprocessing" from the figure, without compromising the presentation of our technical contributions to the core network architecture.
>
> > Q2: It might be beneficial to describe the current state and the challenges more in the main text in addition to the information provided in the Appendix.
>
> Thanks for the suggestion! We have incorporated a new figure (**Figure 6**) in the main text to provide a clearer presentation of our real-world experiments. Due to page limits, a more detailed discussion is provided in the corresponding section in the appendix.

---

> > ### Comment · Reviewer_QGTH · 2023-11-21
> > **Responses to Authors**
> >
> > I appreciate the sincere response from the author. My concerns were addressed in the author's reply.

---

### Official Review · Reviewer_5z4H · 2023-11-02

**Soundness:** 4 excellent
**Presentation:** 3 good
**Contribution:** 3 good
**Rating:** 8
**Confidence:** 3

**Summary:**

In this work, the authors model a probabilistic particle-based fluid simulator-- which they claim to be the first of its kind, to address the problem of learning inaccessible physical parameters of fluids using visual observations such as density, viscosity, pressure, and transferring these visual fluid properties to simulate new scenes with new initial and boundary conditions.

**Strengths:**

- The proposed simulator uses consecutive states/image observations to infer fluid dynamics without complete knowledge of the true physical parameters of the fluid.
- The simulator showcases a direct benefit of inferring latent space components apart from its role in the data-generating process-- namely in terms of intuitive physical inference.
- The authors also show real-world experiments on high-resolution images from dyed-water fluid tanks to estimate fluid positions, a challenging task due to reflection, refraction, and the need for high-fps multi-view images.
- The proposed method can take into account the stochasticity of the underlying fluid dynamics unlike other competitive baselines, and not make fluid category-specific initialization assumptions.

**Weaknesses:**

- I am not sure about the claim of latent intuitive physics being proposed by them first, especially when similar ideas have existed in the literature on causal representation learning and other intuitive physics methods, albeit in different formats. I think this claim can be softened to say this is the first method to perform this for fluids using 3D exemplars.
- The authors report that the initial velocities are zero for initial state estimation-- how does this tally up with the initial conditions in the particle datasets considered? Isn't a random rotation, scaling, and initial velocity applied to the initial fluid body?

**Questions:**

- Is the zero initial state velocity a general assumption for all experiments performed? Does this correspond to $(z_{t=1}, \tilde z_{t=1})$ being zero-initialized?
- Why is there a difference in the type of posterior used in the pre-training and the inference stages, being particle-based and visual, respectively?
- Why is the prior learning less prone to overfitting?

**Details Of Ethics Concerns:**

No ethics concerns.

---

> ### Author Response · Authors · 2023-11-19
> **Responses to Reviewer 5z4H (Part 1)**
>
> Thank you for your comments. We hope that our responses below can adequately address your concerns regarding this paper.
>
> > Q1: I am not sure about the claim of latent intuitive physics being proposed by them first. This claim can be softened to say this is the first method to perform this for fluids using 3D exemplars.
>
> We appreciate your suggestion, and in the revised paper, we have refined the introduction to more precisely highlight the technical contributions. Specifically, the novel aspects of the proposed "latent intuitive physics" approach now include:
> - It presents a novel learning-based approach for fluid simulation, which infers the hidden properties of fluids from 3D exemplars and transfers this knowledge to a fluid simulator.
> - Our approach includes the first probabilistic network for particle-based fluid simulation.
>
> > Q2: The authors report that the initial velocities are zero for initial state estimation-- how does this tally up with the initial conditions in the particle datasets considered?
>
> (1) Clarification:
>
> We only adopt the assumption of zero initial velocities for the **observed visual scene** (Table 2), while applying **non-zero** initial velocities for particle-based experiments (Table 3) and for the novel scenes as well (Table 1, Table 4, and Table 5).
>
> (2) New experiments with non-zero initial velocities of the observed visual scene:
>
> To assess the performance of our method in a more general case, that is, learning from visual observations of fluids with non-zero initial velocities, we modify the training scheme in Stage B by randomly initializing $v_{t=1}^{i=1:N}$ and treating them as trainable parameters. The optimization of $v_{t=1}^{i=1:N}$ is carried out concurrently with the optimization of the visual posteriors $\hat{\mathbf{z}}$.
>
> Below, we evaluate the results against a model trained with true non-zero initial velocities on the observed scene. We here compare
> - Future prediction errors on observed scenes (denoted by Obs.)
> - Average simulation errors on novel scenes with new initial geometries and boundaries (denoted by Geom. and Bdy.):
>
> |            | $\rho=2000,\nu=0.065$ (Obs./Geom./Bdy.) | $\rho=1000,\nu=0.08$ (Obs./Geom./Bdy.)   | $\rho=500,\nu=0.2$ (Obs./Geom./Bdy.)     |
> | ---------- | -|-|-|
> | Trained with true $v_{t=1}^{i=1:N}$ | 39.02 /  33.39 / 39.74 | 39.14 / 33.50 / 38.41 | 43.57 / 38.52 / 47.23 |
> | Trained with unknown $v_{t=1}^{i=1:N}$ | 39.53 / 33.25 / 39.55 | 40.99 / 33.72 / 38.75| 45.31 / 38.34 / 46.90 |
>
> We have included the above results in Appendix E. As we can see, both models produce comparable results, showcasing the ability of our method to infer uncertain initial velocities of fluids by treating them as optimized variables.
>
> > Q3: Is the zero initial state velocity a general assumption for all experiments performed? Does this correspond to $(\mathbf{z}\_{t=1}, \tilde{\mathbf{z}}\_{t=1})$ being zero-initialized?
>
> (1) No, assuming zero initial velocity ($v_{t=1}^{i=1:N}$) is not a universal condition for all experiments. As shown below, we only adopt this assumption for the observed visual scene, while applying non-zero initial velocities for the novel scenes (Table 1, Table 4, and Table 5), as well as for the particle datasets (Table 3).
>
> (2) For the latent states $(\mathbf{z}\_{t}, \tilde{\mathbf{z}}\_{t})$ in all experiments, we do initialize $\mathbf{z}\_{t=1}$ and $\tilde{\mathbf{z}}\_{t=1}$ with zero values, which is similar to the common practice for initializing the hidden states of RNNs. Notably, there is no direct connection between the initialization approaches for latent states and the velocity states.
>
> | Zero-initialized?|Observed visual scenes| Novel scenes |Particle experiments|
> |-|-|-|-|
> | $v_{t=1}^{i=1:N}$| Yes| No| No|
> | $\mathbf{z}\_{t=1}, \tilde{\mathbf{z}}\_{t=1}$ | Yes | Yes| Yes|

---

> ### Author Response · Authors · 2023-11-19
> **Responses to Reviewer 5z4H (Part 2)**
>
> > Q4: Why is there a difference in the type of posterior used in the pre-training and the inference stages, being particle-based and visual, respectively?
>
> As illustrated below, both particle-based posteriors ${\mathbf{z}}\_t$ and visual posteriors $\hat{\mathbf{z}}$ are employed to offer effective training supervision for the prior learner.
> - **Pretraining stage:** The particle-based posterior module $q_\xi(\mathbf{z}\_t \mid \mathbf{x}\_{1: t}, \mathbf{z}\_{t-1})$ conditions on the prediction target $\mathbf{x}\_t$ and the previous particle states along with latent features $(\mathbf{x}\_{1:t-1}, \mathbf{z}\_{1:t-1})$. This setup enables the prior learner $p_\psi(\tilde{\mathbf{z}}\_t \mid \mathbf{x}\_{1:t-1}, \tilde{\mathbf{z}}\_{t-1})$ to generate meaningful latent features that capture hidden properties essential for accurate predictions.
> - **Visual inference stage:** In this phase, true particle states are unavailable. Instead, we rely on visual observations and optimize a new set of variables known as visual posteriors ($\hat{\mathbf{z}}$). These visual posteriors are refined by backpropagating image rendering errors. They serve as adaptation targets for the prior learner, facilitating efficient adaptation to the specific observed fluid.
>
> || Particle-based posterior| Visual posterior|
> | ------------- | -| -|
> | Formulation   | $q_\xi(\mathbf{z}\_t \mid \mathbf{x}\_{1: t}, \mathbf{z}\_{t-1})$                    | $q(\hat{\mathbf{z}} \mid I_{1:T})$|
> | Training stage  | Stage A: Particle-based pretraining                                               | Stage B: Visual inference|
> | Data | Given particle states| Given RGB images when particle states are inaccessible|
> | Optimization      | Optimize the parameters of both the posterior module and the prior module to minimize the KL divergence | Directly optimize $\hat{\mathbf{z}}$ to minimize the image rendering loss |
> | Function |     Facilitate prior learner training when ground truth particle states are available|  Provide adaptation targets for the prior learner in the subsequent Stage C |
>
> > Q5:. Why is the prior learning less prone to overfitting?
>
> In Section 4.3 of the original manuscript, we claimed that "*Instead of finetuning the entire transition network as NeuroFluid (Guan et al., 2022) does, we only finetune the prior learner module, which is more efficient and less prone to overfitting.*" What we intended to convey is that tuning all parameters in the transition model in the visual scene (as NeuroFluid does) can lead to overfitting problems. Due to the unavailability of the ground truth supervision signal in particle space, the transition model may:
> - Forget the pre-learned knowledge of feasible dynamics;
> - Learn physically implausible particle transitions (as the learnable transition network has a wide variety of ways to move the input particles to the expected geometric positions, all of which are sufficient to minimize the image rendering loss).
>
> The evidence is that NeuroFluid shows superior performance compared to our approach for the observed scene with $\rho=500,\nu=0.2$ (as presented in Table 1). However, it underperforms for corresponding novel scenes with new initial and boundary conditions (as detailed in Table 2).
>
> |            | Observed scene, $t \in [51,60]$ | Novel scene, $t \in [2,60]$ (Geom./Bdy.) |
> | ---------- | ------------------------------ | ------------------------------- |
> | NeuroFluid | 33.22                           | 40.88 / 50.73                            |
> | Ours       | 41.15 $\pm$ 0.71                     | 39.03 $\pm$ 0.79 / 47.25 $\pm$ 1.26  |
>
> Unlike NeuroFluid, our approach adapts the physical prior learner to visual scenes, without training a probabilistic physics engine. By leveraging knowledge from the pretraining stage, the transition model is less prone to overfitting on the observed scene. This not only enhances the generalization ability but also significantly reduces the training burden.
>
> We have provided additional clarification on this statement to avoid misunderstanding in the revised paper.

---

### Author Response · Authors · 2023-11-19
**Revision Uploaded**

We thank all reviewers for the constructive comments and have updated our paper accordingly. Please check out the new version!

Specific changes or new results include:
1. Clarify the novel aspects of the "latent intuitive physics" approach and specify its contributions within the context of fluid simulation (Section 1).
2. Refine Figure 3 to involve all network components of the model architecture, including the neural renderer (Section 4).
3. Clarify the overfitting problems of tuning all parameters in the transition model in visual scenes (Section 4.3).
4. Perform multiple simulations of our method to compute the standard deviation for each test sequence (Table 1, 2, 3, and 4 in Section 5).
7. Clarify the evaluation metrics for the experiments (Section 5.1 "Settings", 5.2 and Appendix D.1).
6. Add a schematic diagram (Figure 6) to enhance the presentation of our real-world experiments (Section 6).
7. Add a schematic diagram (Figure 7) illustrating the training process for each stage (Appendix A.1).
8. Add Table 6 to provide a summary of each model component's formulations, inputs, outputs, active training stages, and objectives (Appendix A.1).
9. Add a new experiment demonstrating the method's ability to infer unknown initial velocities of fluids in the observed visual scene (Appendix E).
10. Correct the identified typos.

Please do not hesitate to let us know for any additional comments on the paper.

---

### Author Response · Authors · 2023-11-23

Dear reviewers,

Thank you once again for your time in reviewing our paper. We've made every effort to improve the overall clarity in the rebuttal revision. We would appreciate it if you could consider increasing your rating if you find our responses helpful. If any issues remain, please don't hesitate to inform us.

Best, Authors

---

### Meta-Review · Area_Chair_ux8x · 2023-12-07

**Metareview:**

This submission proposes a probabilistic model capable of inferring parameters of fluids from visual observations and then to transfer these parameters to new scenes. The paper has received four positive scores from the reviewers, who appreciated

- a well designed method,
- performance,
- the general idea of inferring latent space components, and the role of intuitive physics,
- real world experiments on high-resolution images,
- the methods ability to handle stochasticity.

The AC concurs and recommends acceptance.

**Justification For Why Not Higher Score:**

This paper could also be a spotlight. It is interesting, refreshingly new.

**Justification For Why Not Lower Score:**

There is no reason to reject this paper.

---

### Decision · Program_Chairs · 2024-01-16

Accept (poster)